# A tungsten polyoxometalate mediated aqueous redox flow battery with high open-circuit voltage up to 2 V

Weipeng Li[1], Weizhuo Xu[1], Zhaopeng Sun[1], Linning Tang[1], Guohao Xu[1], Xinyue He[1], Yulin Deng[2], Wei Sun[3], Bingjie Zhou[4], Jianfei Song[5] & Wei Liu [1]✉

As a promising stationary energy storage device, aqueous redox flow battery (ARFB) still faces the challenge of low open-circuit voltage, due to the limitation of the potential of water splitting (1.23 V theoretically). Herein, we present a low potential anolyte design by using Na substituted phosphotungstic acid ($3Na\text{-}PW_{12}$) for an aqueous redox flow battery with the high open-circuit voltage up to 2.0 V. The $3Na\text{-}PW_{12}$ can store 5 electrons in the charging process and simultaneously capture $Na^+$ or protons from the dissociation of water, resulting in the increase of electrolyte pH to 11. Because of the high pH value, the hydrogen evolution reaction (HER) is highly suppressed, and the $3Na\text{-}PW_{12}$ is partially degraded into a lacunary structured $PW_{11}$ with extremely low potential down to $-1.1$ V (vs. SHE). After discharging, the captured protons are re-released into the solution, therefore, pH and the structure of $3Na\text{-}PW_{12}$ are recovered. Based on the cyclic pH change and self-regulation process of $3Na\text{-}PW_{12}$ in the charge and discharge process, the aqueous flow battery offered a high-power density of 200 mW cm$^{-2}$ and 160 mW cm$^{-2}$ coupled with $Br_2/Br^-$ and $I_2/I^-$ catholyte respectively.

The highly developed renewable energy resources, such as solar and wind power, raise the demand for stationary energy storage, which is important for enabling stable energy output[1]. The merit of decoupled energy capacity and power output makes aqueous redox flow batteries (ARFBs) a promising candidate for adapting to grid-scale energy storage. In addition, ARFBs are much safer than the current Li-ion battery and have longer lifespan and lower maintenance costs[2,3]. However, the voltage output of ARFBs is low, which is limited to water splitting (theoretical splitting voltage 1.23 V).

A number of redox-active electrolyte solutions have already been developed for ARFBs. The all-vanadium flow battery is one of the most promising ARFBs in commercial applications, which utilizes vanadium ions with different valence states on two sides of the battery. The symmetrical electrolyte design can provide an open-circuit voltage of around 1.5 V and significantly reduce the electrolyte loss caused by ion contamination[4]. Besides vanadium ions, metal electrodes (such as Zn and Fe) are considered active materials because of low electrode potential, mild working conditions and rich reserves in the earth[5-7]. Zn electrode performs well under near neutral conditions and offers a low anodic potential at $-0.75$ V vs. standard hydrogen electrode (SHE). Therefore, aqueous Zinc flow batteries, such as $Zn\text{-}I_2$, $Zn\text{-}Br_2$ and $Zn\text{-}Fe$ batteries, have now achieved a high power density output[5,8,9]. For example, $Zn\text{-}Br_2$ flow battery can provide a voltage over 1.6 V[10]. However, dendrite and dead metal are serious problems for the power output and long-term stability of metal-based flow batteries. The development of all-soluble low potential anolyte is an important strategy to solve the problems of metal flow batteries. One of the approaches is introducing ligands to adjust the properties of the metal

[1]School of Chemistry and Chemical Engineering, Central South University, Changsha, Hunan, PR China. [2]School of Chemical & Biomolecular Engineering and RBI at Georgia Tech, Georgia Institute of Technology, Atlanta, GA, USA. [3]School of Chemistry and Chemical Engineering, Shihezi University, Shihezi, PR China. [4]National Engineering Laboratory for Mobile Source Emission Control Technology, China Automotive Technology & Research Center Co. Ltd., Tianjin, PR China. [5]Changsha New Energy Innovation Institute, Changsha, Hunan, PR China. ✉e-mail: wliu300@csu.edu.cn

redox pairs[11,12]. Gong et al. used TEOA as a ligand, which decreased the potential of $Fe^{3+}/Fe^{2+}$ from 0.77 V to −0.84 V vs. SHE, but the organic ligand could be potentially degraded in the charge and discharge process[13].

Polyoxometalates (POMs), which are a kind of soluble metal cluster anions, have great potential applied in ARFBs[14]. POMs are mostly composed of heteroatom (X: Si, P, Zn, etc.) and framework atoms (M: Mo, W, V, etc.) to form Keggin ($XM_{12}$) or Dawson ($X_2M_{18}$) type structures[15]. For example, Dawson structured $P_2W_{18}$ (the lowest potential: about −0.5 V vs. SHE) can store 18 electrons and exhibit remarkable redox activity and high solubility even at −20 °C in the charge and discharge of ARFBs, which benefits from its electron delocalization property and strong $H^+$ solvation shell structure[16,17]. Due to the high structural flexibility, the properties of POMs can be tuned by changing the heteroatom or framework atom to fabricate multi-substituted POMs[18,19]. Replacing W atoms with V atoms in $SiW_{12}$, $SiW_9V_3$ can be obtained and has been used as a bipolar electrolyte in ARFBs[20]. By changing the heteroatom to form $CoW_{12}$, it can provide a 4-electron reversible redox reaction of framework atoms and a high redox potential (1.0 V vs. SHE) due to the high valence of the center atom Co[21]. Taking advantage of proton coupled electron transfer (PCET) mechanism, the symmetrical $CoW_{12}$ flow battery can provide a voltage of 1.5 V by adjusting the pH value of the electrolyte[22]. Large tungsten POMs (such as $P_5W_{30}$ and $P_8W_{48}$) can store over 20 electrons in ARFBs but they need to be overcharged, which leads to low columbic efficiency in ARFBs because the problem of hydrogen evolution is still inevitable[23]. Therefore, the performances of ARFBs are still required to be further improved for renewable electric energy storage.

Herein, we presented an aqueous polyoxometalate flow battery with low anodic potentials design, which can provide the highest open-circuit voltage reached up to 2.0 V with a coupled halogen catholyte (such as $Br_2/Br^-$ or $I_2/I^-$). We found a self-regulation phenomenon of 3 sodium substituted phosphotungstic acid (3Na-$PW_{12}$) in the charge/discharge process that 3Na-$PW_{12}$ can reversibly receive/release 5 electrons and simultaneously capture/release protons from water

dissociation, resulting in the cyclic changes of electrolyte pH between 1.3 and 11. Notably, it is because the dynamic pH changes to a high value, which suppresses the hydrogen evolution reaction, that the 3Na-$PW_{12}$ can be highly reduced and reach an low potential of −1.1 V vs. SHE. The potential obtained by reduced 3Na-$PW_{12}$ is lower than most of the metal electrodes and reported POMs materials as far as we know, indicating that high open-circuit voltage outputs of polyoxometalate flow batteries could be obtained by using it as anolyte. Meanwhile, $SiW_{12}$ ($H_4SiW_{12}O_{40}$) exhibits a similar structure and self-regulation phenomenon to $PW_{12}$, implying the significant application potential of tungsten-based polyoxometalates in high performance aqueous redox flow batteries.

## Results

### Electrochemical behavior of 3Na-$PW_{12}$

As shown in the aqueous polyoxometalate flow battery (Fig. 1a), 3Na-$PW_{12}$ was introduced as anolyte and $Br^-/Br_2$ was applied as catholyte. The 3Na-$PW_{12}$ anolyte was prepared via the gradual replacement of protons in Keggin-structured phosphotungstic acid with sodium ions, as detailed in electrolyte preparation. When the battery was charging, the $PW_{12}^{3-}$ anions received electrons and spontaneously captured sodium ions and protons from the dissociation of water, which leads to the increase of the pH of the anolyte. After fully charging, the captured protons could be gradually released back into the anolyte during the discharging process, verified by Fig. 1b. Figure 1b shows the electron numbers received by each $PW_{12}^{3-}$ anion with the change of pH in the charging and discharging process. The pH value could reach 11 after the charging, meanwhile, $PW_{12}^{3-}$ anion received 5 electrons. With the increase of pH value to 11, the standard electrode potential for hydrogen evolution reaction (HER) decreases to −0.65 V (Vs. NHE) according to the Nernst equation, which means the water splitting reaction was suppressed. Meanwhile, there is a significant kinetic overpotential on the simple graphite electrode. Therefore, the redox potential of 3Na-$PW_{12}$ anolyte could be dropped to −1.1 V vs. SHE without hydrogen evolution. This value is far lower than common

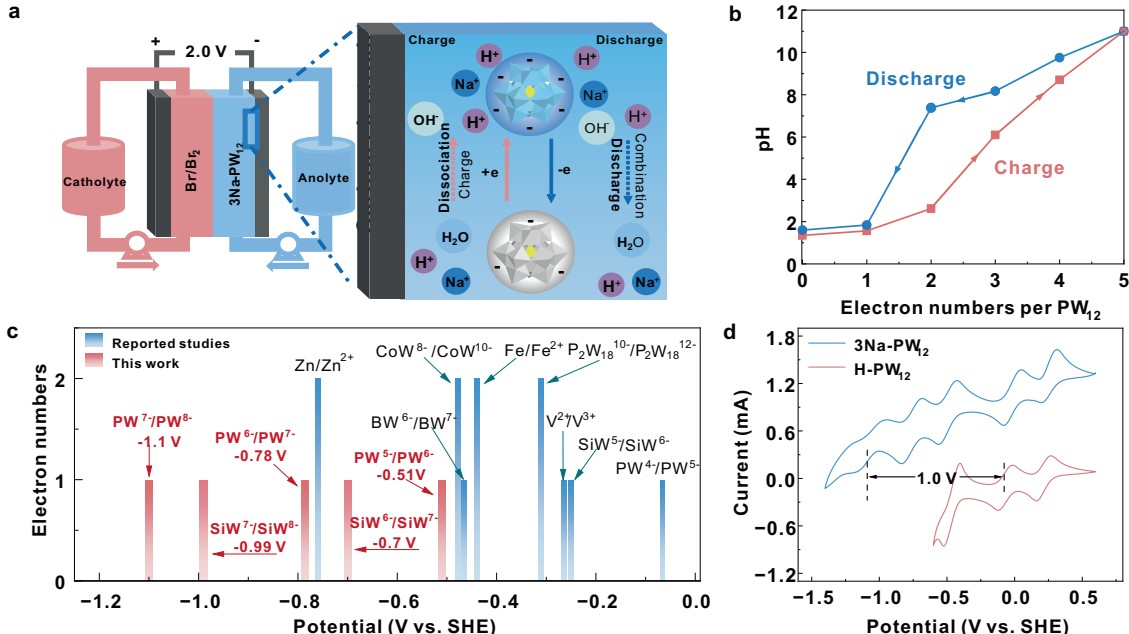

**Fig. 1 | Electrochemical behaviors of 3 sodium substituted phosphotungstic acid (3Na-$PW_{12}$) in ARFB. a** Schematic diagram of the 3Na-$PW_{12}$ redox flow battery. **b** The cyclic pH change of 3Na-$PW_{12}$ anolyte (0.1 mol l$^{-1}$) in a complete charge (red line) and discharge (blue line) process, obtained by in-situ pH monitoring with 1 mol l$^{-1}$ NaI as catholyte. **c** Redox potentials of redox-active materials presented in the reported and this study ($PW_{12}$ and $SiW_{12}$ are the abbreviation of Keggin type POMs $H_3PW_{12}O_{40}$ and $H_4SiW_{12}O_{40}$ respectively; $P_2W_{18}$ is the abbreviation of $H_6P_2W_{18}O_{60}$)[16,22,24,36,37]. **d** Cyclic voltammetry (CV) curves of the charged 0.1 mol l$^{-1}$ H-$PW_{12}$ and 3Na-$PW_{12}$ anolyte with a saturated Ag/AgCl electrode and the scan rate of 100 mV s$^{-1}$ at room temperature without iR-correction.

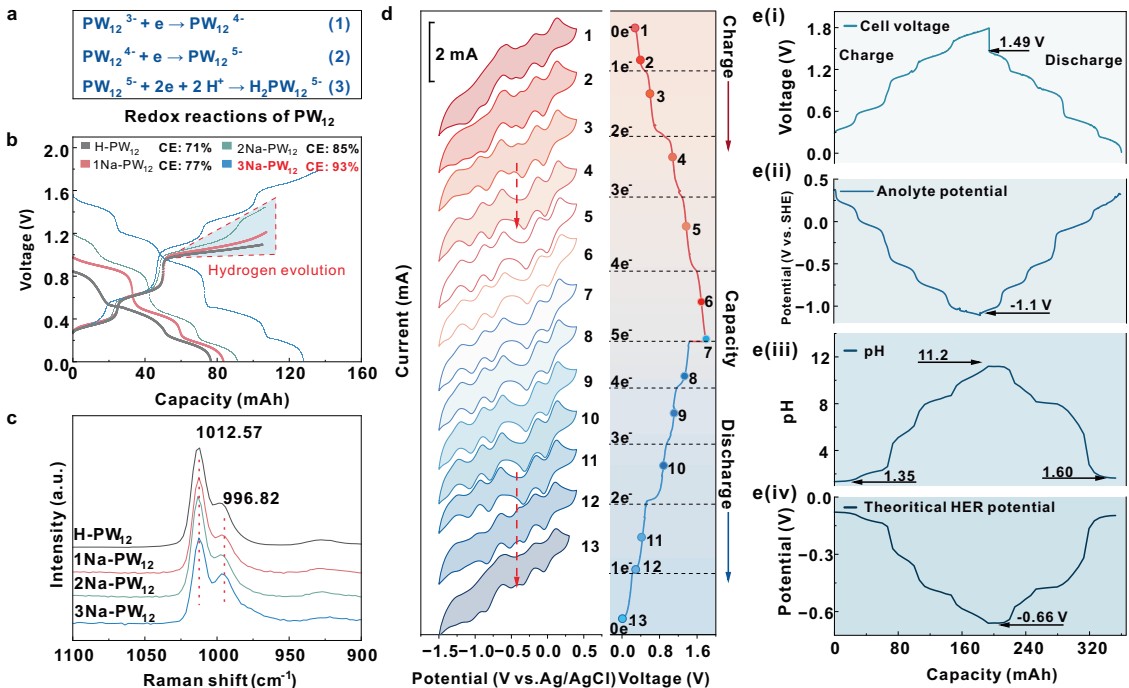

**Fig. 2 | Characteristics and in-situ monitoring of 3Na-PW₁₂ anolyte in charge and discharge process. a** Redox reactions of H-PW₁₂. **b** Galvanostatic charge and discharge (GCD) curves of H, 1Na, 2Na and 3Na-PW₁₂ as anolyte (15 ml, 0.1 mol l⁻¹) with NaI (15 ml, 1 mol l⁻¹) as catholyte at 25 mA cm⁻². **c** Raman spectra of H, 1Na, 2Na and 3Na-PW₁₂ (0.1 mol l⁻¹). The unit 'a. u.' stands for 'arbitrary unit'. **d** CV in-situ monitoring of 3Na-PW₁₂ (0.1 mol l⁻¹) during GCD process at 25 mA cm⁻² with a

saturated Ag/AgCl electrode and the scan rate of 100 mV s⁻¹ at room temperature without iR-correction. **e** In-situ potential and pH monitoring of 3Na-PW₁₂ (0.1 mol l⁻¹) during the GCD process: (i) GCD curves at 25 mA cm⁻², (ii) potentials of 3Na-PW₁₂ anolyte, (iii) pHs and (iv) theoretical HER potentials. Conditions for GCD curves measurements: electrolyte solution flow rate: 90 ml min⁻¹, at room temperature, without iR-correction.

redox pairs applied in ARFBs, as shown in Fig. 1c, Supplementary Table 1 and Supplementary Fig. 1. For example, the redox potential of the V²⁺/V³⁺ pair presented in the anodic side of the all-vanadium flow battery is −0.27 V vs. SHE in 3 mol l⁻¹ H₂SO₄ solution. The metal anodes, such as Fe and Zn, can offer −0.42 V and −0.75 V vs. SHE respectively in near neutral solution. The POMs with the lowest fpotential reported as far as we know is CoW₁₂, which shows a 2 electrons reversible potential at −0.48 V vs. SHE (pH = 4.0). In this work, 3Na-PW₁₂ exhibited low potential down to −1.1 V vs. SHE (pH = 11) but the acid form of PW₁₂³⁻ ions (H-PW₁₂, pH < 1) only provides the negative potential at −0.07 V. This can be further certificate by CV curves of H-PW₁₂ (Fig. 1d), which shows characteristic three redox peaks (0.21 V, −0.07 V and −0.50 V vs. SHE) of H-PW₁₂. However, 3Na-PW₁₂ shows five pairs of redox peaks centered at 0.21 V, −0.07 V, −0.51 V, −0.78 V and −1.1 V vs. SHE. The newly added redox peaks indicate the low potential property of 3Na-PW₁₂.

### In-situ electrochemical characteristics during the galvanostatic charge/discharge (GCD) process

It is well known that the typic PW₁₂³⁻ anion receives the first two electrons through single-electron transfer reactions, and then follows with a protons-coupled two electrons transfer (PCET)[24], as shown in Fig. 2a. These three reactions are respectively corresponding to the three redox peaks located at 0.21 V, −0.07 V and −0.5 V vs. SHE in the CV curve of H-PW₁₂ (Fig.1d). However, only the first two single-electron transfers are reversible in aqueous solution. The PCET reaction is commonly accompanied by HER in acidic solution because the PCET potential drops to −0.5 V vs. SHE[24]. Therefore, 1Na, 2Na and 3Na substituted PW₁₂ were prepared and employed in the aqueous flow battery for the purpose of HER mitigation. Figure 2b shows the galvanostatic charge/discharge (GCD) curves of the flow battery with different Na substituted PW₁₂ solution

as anolyte and NaI solution as catholyte. The coulombic efficiency (CE) of assembled flow battery increased with the raise of Na⁺ substitution numbers and the corresponding decrease of proton concentrations. These results suggest that Na⁺ substituted PW₁₂ can store more electrons than the acid form of PW₁₂ (H-PW₁₂). 3Na-PW₁₂ can even store 5 electrons and provide the highest open-circuit voltage of almost 1.5 V with a high coulombic efficiency of 93% but the H, 1Na and 2Na-PW₁₂ show obvious hydrogen evolution, which was also confirmed with CV curves of different Na substituted PW₁₂ scanned at a broad potential window (Supplementary Fig. 2).

By continuously adding Na⁺ into the H-PW₁₂ solution with the substitution number higher than 3, however, the Keggin structure will be decomposed. Polyoxometalates are commonly stable in mildly acidic solution because H⁺ ions interact with the surface oxygen atoms of the PW₁₂, which reduces the negative charge density of frameworks. However, the degradation becomes remarkably feasible in alkaline media by OH⁻ attacking of W-O-W bonds and leading to subsequent structural decomposition of polyoxometalate frameworks (shown in Supplementary Fig. 6)[15,25,26]. The final decomposed products are WO₄²⁻ and PO₄³⁻ at a high pH condition and lose the redox activity on the graphite electrode, verified by CV curves of PW₁₂ under a higher molar ratio of Na⁺ to PW₁₂ (Supplementary Fig. 3). The 1Na, 2Na and 3Na substituted PW₁₂ are stable and can maintain the Keggin structure, which was confirmed with Raman (Fig. 2c) and UV-Vis (Supplementary Fig. 4) spectra. Raman peak at 996 cm⁻¹ results from the asymmetric vibration of W-O_d. The symmetric stretching vibration of W-O_d is located at the peak at 1012 cm⁻¹, which is considered to be the identification peak of the Keggin structure of PW₁₂[27]. The UV-Vis absorption peaks around 206 nm and 265 nm were caused by the charge excitation of O_d → W and O_b, O_c → W respectively (Supplementary Fig. 4)[28]. In addition, the Raman and CV measurements were performed with high numbers of Na⁺ substitutions from 4 to 9 of PW₁₂ (Supplementary

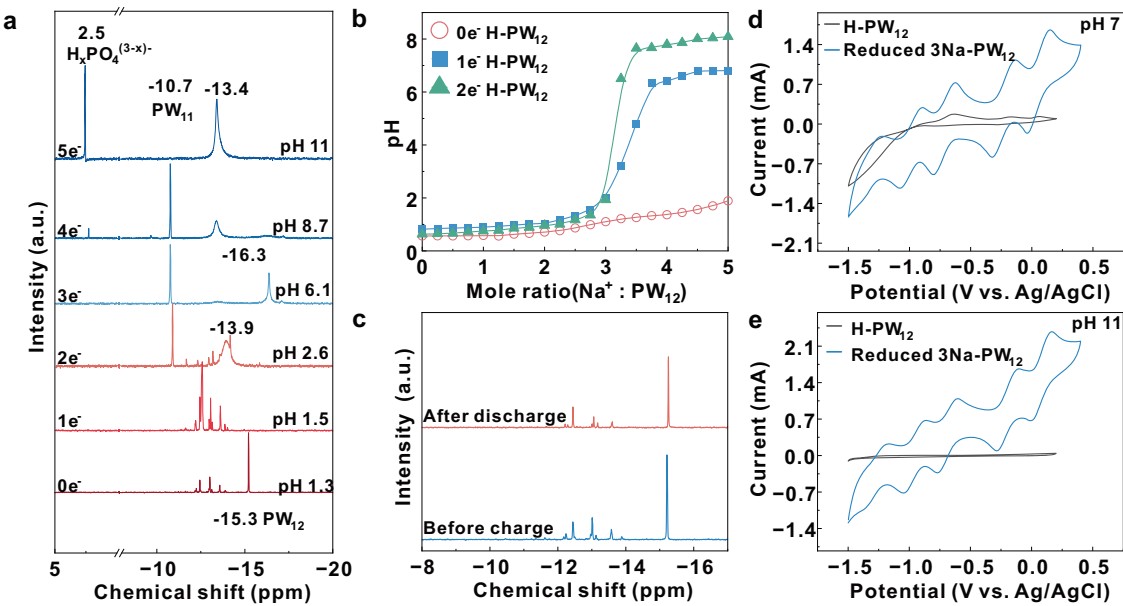

**Fig. 3 | Structural and electro-active investigations. a** $^{31}$P NMR of different reduced states of 3Na-PW$_{12}$ (0.1 mol l$^{-1}$) anolyte in the charging process. **b** The acid-base titration curves of different reduced states of H-PW$_{12}$ (room temperature). **c** $^{31}$P NMR of 3Na-PW$_{12}$ anolyte (0.1 mol l$^{-1}$) before charging (blue line) and after discharging (red line) at pH 1.3. **d** CV curves of H-PW$_{12}$ and 3Na-PW$_{12}$ (0.1 mol l$^{-1}$) at pH 7 and (**e**) at pH 11 (scan rate 100 mV s$^{-1}$, room temperature without iR-correction).

Fig. 5). However, the Raman peak at 1012 cm$^{-1}$ decreased, indicating the decomposition of PW$_{12}$[29].

To investigate the changes of PW$_{12}$ anion in the galvanostatic charge/discharge (GCD) process, in-situ measurements of the CV curves, potentials and pH values of 3Na-PW$_{12}$ anolyte solution under different states of charge were performed (Fig. 2d, e). During the charge process, the CV curves have no obvious change at the first two charge platforms (CV number: from 1 to 3). When it was charged to the third platform, which fully stored 2 electrons for each PW$_{12}$ (CV number: 4), the redox peak centered around −0.45 V disappeared, and expectedly, the CV curves turned into five pairs of distinct redox peaks. Notably, the fifth peak centers around −1.1 V vs. SHE, which indicates the theoretical open-circuit voltage could be up to 1.64 V coupled with the I$^-$/I$_2$ catholyte. In the discharging process, the shape of CV curves was maintained in the first four discharge platforms (CV numbers: from 7 to 11), and finally returned to the original shape (CV numbers: 12–13). In a complete GCD process, each PW$_{12}$ reversibly stored and released 5 electrons without obvious hydrogen evolution. The actual open-circuit voltage reached 1.5 V after the charging process and the lowest anolyte potential was dropped to −1.1 V vs. SHE, as shown in Fig. 2e (i and ii). The pH changed (Fig. 2e(iii)) slightly at the first two charging platforms, then raised rapidly in the following three platforms, and eventually reached the pH value of 11 when fully charged. Based on the Nerst equation(as shown in Eq. 1), we calculated the theoretical HER potentials with the changes in pH values (Fig. 2e(iv)). The theoretical HER potentials could be decreased to −0.66 V vs. SHE in the charging process, indicating that the water splitting was highly suppressed.

## Structure changes of 3NaPW$_{12}$ in the ARFB anolyte solution

To understand the Keggin structure changes of PW$_{12}$ during the GCD process, phosphorus nuclear magnetic resonance ($^{31}$P NMR) was used to investigate the species presented in the PW$_{12}$ solutions with different reduced states. As shown in Fig. 3a, 0e$^-$ represents the uncharged 3Na-PW$_{12}$ solution and the peak at −15.3 ppm corresponds to PW$_{12}$. There are small peaks between −11 and −13 ppm, which means the partial decomposition of the initial 3Na-PW$_{12}$ electrolyte[30]. With reduced to 2e$^-$ state, the new peak at −10.7 ppm appeared, which could be ascribed to the formation of lacunary Keggin structure PW$_{11}$O$_{39}$$^{7-}$

(PW$_{11}$). The signal of the PW$_{11}$ peak continues to strengthen with further reduction. Upon reducing to the 4e$^-$ state, the peak of H$_x$PO$_4$$^{(3-x)-}$ (at 2.5 ppm) begins to appear, accompanied by the weakening of the peak of PW$_{11}$. At the final reduction of 5e$^-$ state, the peak of PW$_{11}$ at −10.7 ppm shifted to −13.4 ppm possibly because it was highly reduced. The NMR analysis indicates that reduced PW$_{12}$ was partially degraded into PW$_{11}$ in the anolyte, but it still maintains redox activity even at a high pH value.

The stability of initial and reduced PW$_{12}$ under a high alkaline condition was investigated by the titration experiments. Figure 3b shows the acid-base titration curves of H-PW$_{12}$ with different reduced states. NaOH was added into the initial state H-PW$_{12}$ (0e$^-$ H-PW$_{12}$) until the molar ratio of Na$^+$:PW$_{12}$ reached 5. The pH value was still lower than 2 because PW$_{12}$ was gradually decomposed into WO$_4$$^{2-}$ and PO$_4$$^{3-}$, and thus consumed the newly added OH$^-$ (shown in Supplementary Fig. 6). When neutralizing the reduced PW$_{12}$ (1e$^-$ and 2e$^-$ H-PW$_{12}$), the pH jumped rapidly at the Na$^+$:PW$_{12}$ ratio of 3, which indicates that reduced PW$_{12}$ or PW$_{11}$ cannot consume the excess OH$^-$. Therefore, the stability of tungsten based polyoxometalates in the high pH solution was highly improved after it was reduced. During the charging process, with the pH increased to 11, reduced PW$_{12}$ was partially degraded into PW$_{11}$, but it will not be continuously decomposed into WO$_4$$^{2-}$ and PO$_4$$^{3-}$. As the pH changed back to initial value in the discharge process, PW$_{11}$ and decomposed species of PW$_{12}$ would self-assemble back into PW$_{12}$, which is recognized as the self-healing property of polyoxometalate (as shown in Supplementary Fig. 6). This was also verified by the $^{31}$P NMR of the 3Na-PW$_{12}$ anolyte solutions before charging and after discharging (Fig. 3c). The two spectra are almost the same, indicating that the anolyte solution after discharging was basically restored to the initial state of PW$_{12}$. Species such as WO$_4$$^{2-}$ produced by the decomposition of PW$_{12}$ at high pH usually have very weak redox activities, as evidenced by the CV curves in Fig. 3d, e. However, the reduced 3Na-PW$_{12}$ was able to maintain stable and exhibited remarkable redox activity under neutral (pH 7), or even high alkaline conditions (pH 11).

## Theoretical calculations and self-regulation mechanism

To further understand the self-regulation mechanism of 3Na-PW$_{12}$ electrolyte, density-functional theory (DFT) calculations and

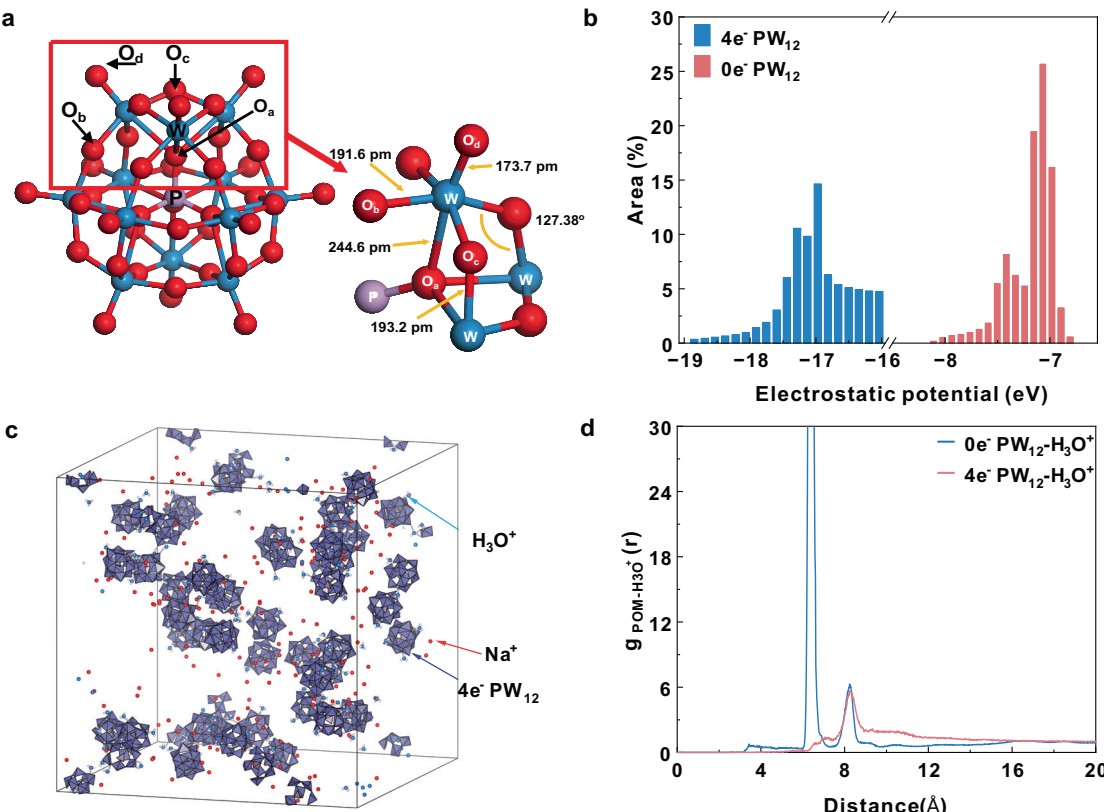

**Fig. 4 | Density-functional theory (DFT) calculations and molecular dynamics (MD) simulations. a** Structure and the subunit of $4e^-$ reduced $PW_{12}$. **b** Surface area in each electron static potential (ESP) range on the $0e^-$ $PW_{12}$ and $4e^-$ $PW_{12}$ surface. **c** snapshot of a representative 3D-periodic simulation box for $4e^-$ $PW_{12}$ in MD simulations (water molecules were removed). **d** POM·$H_3O^+$ Radial Distribution Functions (RDFs) calculated from classical MD simulations using the center of mass of each $PW_{12}$ as reference.

molecular dynamics (MD) simulations were performed with $0e^-$ $PW_{12}$ and $4e^-$ reduced $PW_{12}$ molecules respectively. GROMACS 2024.0 with the Amber 03 force field and the ChelpG charges was applied for MD simulation. Calculations were conducted in Gaussian 16 at the DFT/PBE0 level with the LANL2DZ (ECP) basis set. Solvent effects were included via the integral equation formalism polarizable continuum model (IEF-PCM) during geometry optimization. The Keggin structural $PW_{12}$ is composed of four trimetallic oxide clusters ($W_3O_{13}$) and a central heteroatom atom (as shown in Fig. 4a). The oxygen in $PW_{12}$ can be classified into four types: the $O_a$ residing in the P-O tetrahedron, the bridge connected $O_b$ between two different $W_3O_{13}$ triplets, the bridge connected $O_c$ in the same $W_3O_{13}$ group, and the terminal oxygen $O_d$ (Fig. 4a). After the $4e^-$ reduction of $PW_{12}$, the bond lengths of all W-O bonds become longer than initial $PW_{12}$, and the W-$O_d$ bond has the largest length increase, which reached 4 pm, indicating slightly distortion of the Keggin structure (Supplementary Fig. 7). The electrostatic potential (ESP) analysis was evaluated by Multiwfn according to an efficient algorithm proposed in the literature[31,32]. Owing to the electron delocalization properties, the structure of $4e^-$ reduced $PW_{12}$ can remain stable even under the high pH value of 11. The surface area ESP distribution in the $PW_{12}$ molecule is shown in Fig. 4b. The $4e^-$ reduced $PW_{12}$ exhibits a more negative ESP ranging from −19 to −16 eV compared to $0e^-$ $PW_{12}$. The negative charge is mainly distributed on the $O_c$ and $O_b$ atoms (Supplementary Fig. 8). Furthermore, we evaluated the interaction between the highly negative charged $PW_{12}$ and the cations in the electrolyte by MD simulations. Figure 4c shows a typical snapshot of a representative 3D-periodic simulation box for $4e^-$ $PW_{12}$ MD simulation. The snapshot of the simulation box for $0e^-$ $PW_{12}$ is shown in Supplementary Fig. 9. The radial distribution functions (RDFs) of $0e^-$ $PW_{12}$ and $4e^-$ $PW_{12}$ with $H_3O^+$ are shown in Fig. 4d. The

results indicate that $H_3O^+$ has stronger interactions with $4e^-$ $PW_{12}$ compared to $0e^-$ $PW_{12}$, due to the high negative charge of $4e^-$ $PW_{12}$ (shown in Supplementary Fig. 10a, b). The distance between $4e^-$-$PW_{12}$ anions is longer than that of $0e^-$-$PW_{12}$ anions (Supplementary Fig. 10c), because of stronger interaction between cations ($H_3O^+$ or $Na^+$) and $4e^-$-$PW_{12}$ (Supplementary Fig. 10d). $Na^+$ cation also strongly interacts with $4e^-$-$PW_{12}$ according to the $4e^-$ $PW_{12}$-$Na^+$ RDFs (as shown in Supplementary Fig. 11). It is shown that the reduced $PW_{12}$ is attractive to cations including both $H_3O^+$ and $Na^+$ in the electrolyte solution.

### Electrochemical kinetics studies and electrochemical performances of 3Na-$PW_{12}$ in ARFB

The charging/discharging kinetics of 3Na-$PW_{12}$ anolyte on the surface of the graphite electrode were investigated in detail. Figure 5a shows the CV curves of 3Na-$PW_{12}$ anolyte with different scan rates. According to the Randles-Sevcik equation (the theoretical model for diffusion-controlled systems), the peak currents ($i_p$) were linearly fitted with the square root of the scan rate ($v$) (Supplementary Fig. 12), which implies that the charge and discharge reactions of 3Na-$PW_{12}$ are controlled by diffusion. Supplementary Fig. 13a shows the CV curves of H-PW12 at different scan rate. The diffusion coefficients of 3Na-$PW_{12}$ and H-$PW_{12}$ in the electrolyte solution were measured by the rotating disk electrode (RDE) method. Figure 5b and Supplementary Fig. 13b shows the linear sweep voltametric (LSV) curves of 3Na-$PW_{12}$ and H-$PW_{12}$ at different rotation speeds from 50 rpm to 1200 rpm respectively. The Levich equation (Fig. 5c and Supplementary Fig. 13c) was used to calculate the diffusion coefficient of 3Na-$PW_{12}$ ($1.7 \times 10^{-7} cm^2 s^{-1}$) and H-$PW_{12}$ ($2.4 \times 10^{-7} cm^2 s^{-1}$) (Supplementary Fig. 13f). Furthermore, the kinetic rate constants of 3Na-$PW_{12}$ ($4.6 \times 10^{-4} cm s^{-1}$) and H-$PW_{12}$ ($6.7 \times 10^{-4} cm s^{-1}$) were determined by using the Butler-Volmer and

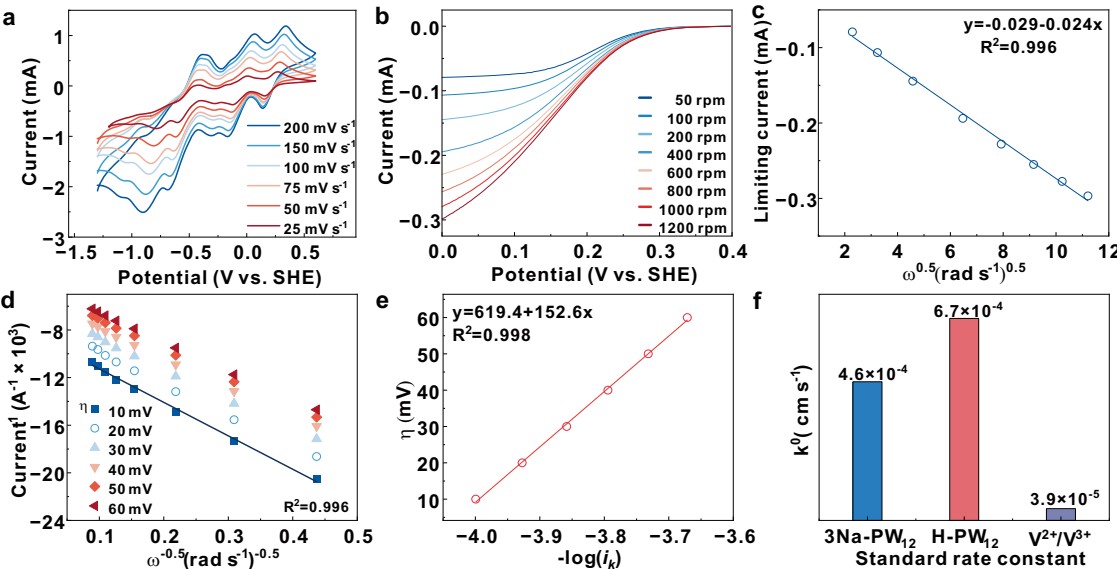

**Fig. 5 | Electrochemical kinetics of 3Na-PW$_{12}$. a** CV curves of 3Na-PW$_{12}$ (0.1 mol l$^{-1}$) with different scan rates (without iR-correction). **b** Rotating disk electrode (RDE) measurements with different rotation speeds (without iR-correction). **c** Levich plot, (**d**) Koutecky–Levich plots, (**e**) Butler–Volmer plot of 0.01 mol l$^{-1}$ 3Na-PW$_{12}$. $\eta$ is overpotential, $\omega$ is rotation rate and $i_k$ is the kinetic current calculated according to the Koutecky-Levich equation. **f** Standard rate constants $k^0$ of 0.01 mol l$^{-1}$ 3Na-PW$_{12}$, H-PW$_{12}$ and V$^{2+}$/V$^{3+}$. CV and RDE tests were conducted under RT.

Koutecky-Levich equations (Figs. 5d, e and Supplementary Fig. 13d, e). It can be seen that the kinetic rate constant of PW$_{12}$ anolyte is 10 times higher than that of V$^{2+}$/V$^{3+}$ redox couple which is the anode side of commercial all-vanadium flow battery (Fig. 5f)[33]. This is because PW$_{12}$ exhibits lower reorganization energy ($\lambda$) compared to V$^{2+}$/V$^{3+}$, due to its structural rigidity (suppressing geometric distortion) and large size (reducing solvent reorganization). At the same time, electron-delocalized polyoxometalate framework of PW$_{12}$ facilitates efficient electron transfer pathways, resulting in higher reaction rates[22,26,34].

The performances of 3Na-PW$_{12}$ in ARFBs were investigated coupled with I$_2$/NaI or Br$_2$/NaBr as the catholyte in the battery tests (Fig. 6a). Nafion 117 membrane and 9 cm$^2$ or 16 cm$^2$ graphite felt (with thickness 2 mm) electrode were used in the flow battery. The ohmic resistance of the flow batteries were determined by the EIS measurements, showing that the ohmic resistances of the flow battery with 3Na-PW$_{12}$ (0.1 mol l$^{-1}$)-NaI (1 mol l$^{-1}$) and 3Na-PW$_{12}$ (0.3 mol l$^{-1}$)-NaI (2 mol l$^{-1}$) as electrolyte solution are 2.69 ± 0.041 Ω and 2.59 ± 0.017 Ω, respectively (Supplementary Fig. 14). The Br$_2$/Br$^-$ couple has a high redox potential (-1.07 V vs. SHE) and was selected in the testing of the flow battery to achieve a high open-circuit voltage. The anodic electrolyte was sealed in an N$_2$ environment to prevent oxidation by atmospheric oxygen. As shown in Fig. 6b, the 3Na-PW$_{12}$ (0.3 mol l$^{-1}$)-Br$_2$ (2 mol l$^{-1}$) flow battery achieved an initial discharge voltage of 1.85 V at 25 mA cm$^{-2}$ with energy density of 36.5 Wh l$^{-1}$ at room temperature without iR-correction. The volumetric capacity of the 3Na-PW$_{12}$ anolyte (0.3 mol l$^{-1}$) is 32 Ah l$^{-1}$ at the first GCD cycle, which is approximately 2.5 times of the H-PW$_{12}$ anolyte (Supplementary Fig. 15). This result indicates that nearly 5 electrons are stored for each PW$_{12}$ and the atomic utilization efficiency (the ratio of tungsten atoms evolved in the redox reactions) is as high as 42%. However, the atomic efficiency of H-PW$_{12}$ is only 16%. As far as we know, the atomic utilization of 3Na-PW$_{12}$ is higher than most of the reported POMs applied in ARFBs (Supplementary Table 2). The 3Na-PW$_{12}$ (0.3 mol l$^{-1}$)-Br$_2$ (2 mol l$^{-1}$) flow battery can present a high open-circuit voltage up to 2.0 V, with a maximum power density of 200 mW cm$^{-2}$ at 200 mA cm$^{-2}$ discharging current density without iR-correction (Fig. 6c). The 3Na-PW$_{12}$-I$_2$ flow battery had a maximum power density of 160 mW cm$^{-2}$, which is still 1.8 times of the H-PW$_{12}$-I$_2$ flow battery (58 mW cm$^{-2}$, Supplementary Fig. 16). The

energy density of 3Na-PW$_{12}$-I$_2$ (25 Wh l$^{-1}$) also improved by more than 4 times that of H-PW$_{12}$-I$_2$ (4.8 Wh l$^{-1}$). Rate performances of the 0.1 mol l$^{-1}$ 3Na-PW$_{12}$-I$_2$ flow battery (charge cut-off voltage: 1.6 V) are given in Fig. 6d, which shows a high value of 8.3 Ah l$^{-1}$ at 20 mA cm$^{-2}$ and 2.4 Ah l$^{-1}$ at 100 mA cm$^{-2}$ (0.1 mol l$^{-1}$). Figure 6e shows the corresponding GCD curves of rate performance, the charge voltage gets higher as the increase of current density. Cycle performance was measured at 50 mA cm$^{-2}$, with 1.7 V and 0.4 V chosen as the charge and discharge cut-off voltages respectively, as shown in Supplementary Fig. 17. Figure 6f shows the calculated capacity, coulombic efficiency (CE) and energy efficiency (EE) in corresponding cyclic tests of the flow battery with 3Na-PW$_{12}$ (0.1 mol l$^{-1}$)-NaI (1 mol l$^{-1}$), indicating the stable operation of 40 cycles, with an average CE of 98.3% (with standard deviation (SD) of 1.52), an average EE of 60% (with SD of 1.97) (Supplementary Table 3) and capacity decay rate of 0.12% per cycle. In order to verify the reproducibility of the aqueous flow battery, a flow battery with the electrode area of 16 cm$^2$ was duplicated, and it shows similar electrochemical performance under the same measurement conditions. In the 120 cycles of GCD tests on the duplicated flow battery, an average CE of 98.5% (with SD of 0.53), an average EE of 67% (with SD of 1.52), and capacity decay of 0.09% per cycle were obtained, indicating good reproducibility of the NaPW$_{12}$-NaI flow battery (Supplementary Fig. 18). The solubility of the 3Na-PW$_{12}$ anolyte was measured to be 0.38 mol l$^{-1}$ (1119 g l$^{-1}$) at 25 °C and 0.45 mol l$^{-1}$ (1325 g l$^{-1}$) at 35 °C, which means that the theoretical capacities of 50.9 Ah l$^{-1}$ and 60.3 Ah l$^{-1}$, respectively, of the anolyte could be reached (Supplementary Fig. 19).

In addition, a similar study was performed with Na-substituted H$_4$SiW$_{12}$O$_{40}$ (SiW$_{12}$) as anolyte. UV-Vis spectra of different Na substitutions of SiW$_{12}$ also show similar absorption peaks to that of PW$_{12}$ at 205 and 258 nm, which could be ascribed to charge excitation of W-Od and W-Ob,c bonds of SiW$_{12}$, respectively (Supplementary Fig. 20a)[26]. Raman spectra show similar peaks of 0-4 Na substituted SiW$_{12}$ at 999 cm$^{-1}$ due to the asymmetric vibration of W-Od (Supplementary Fig. 20b)[27,28]. These spectroscopic characterizations indicate that replacing the protons with 4 sodium ions has no significant structural changes to the SiW$_{12}$ frameworks. The CV curve of 2e$^-$ reduced 4Na-SiW$_{12}$ shows four redox peaks, which are corresponding to four single electron transfer process[35], and the lowest redox potential is decreased

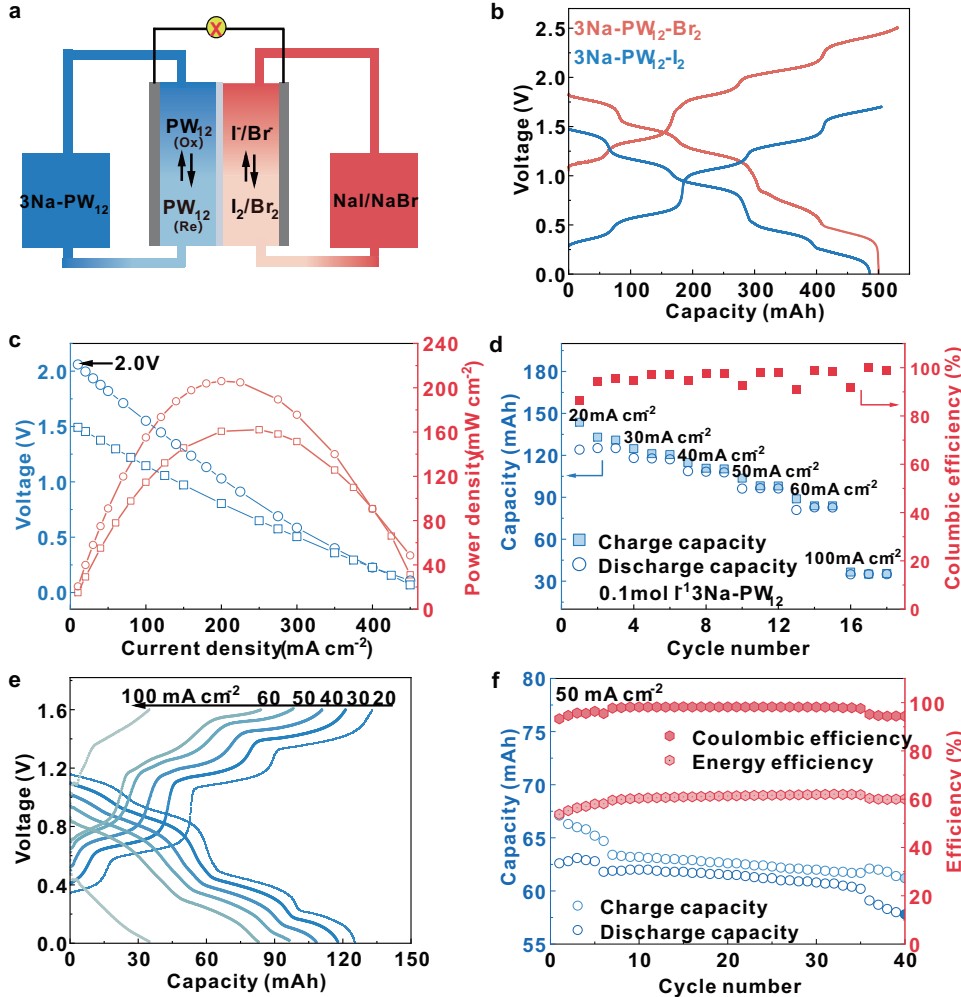

**Fig. 6 | Electrochemical performance of 3Na-PW$_{12}$ flow batteries. a** The illustration of the assembled flow battery. **b** GCD curves of 3Na-PW$_{12}$ (15 ml, 0.3 mol l$^{-1}$)-NaI and NaBr (15 ml, 2 mol l$^{-1}$) at 25 mA cm$^{-2}$. **c** Polarization plots of the aqueous flow battery with 3Na PW$_{12}$ (0.3 mol l$^{-1}$)-NaI and NaBr (2 mol l$^{-1}$). **d** Rate performances of 3Na PW$_{12}$ (0.1 mol l$^{-1}$)-NaI (1 mol l$^{-1}$) at different current densities and (**f**) Capacity of charging (light blue) and discharging (dark blue), coulombic efficiency (CE, dark red) and energy efficiency (EE, light red) in cyclic tests at 50 mA cm$^{-2}$ of the flow battery with 3NaPW$_{12}$ (0.1 mol l$^{-1}$)-NaI (1 mol l$^{-1}$). Conditions for GCD and polarization curves measurements: electrolyte solution flow rate: 90 ml min$^{-1}$, at room temperature, without iR-correction.

to −1.19 V (vs. Ag/AgCl), as shown in Supplementary Fig. 21. The 4Na-SiW$_{12}$-I$_2$ flow battery exhibited four voltage platforms and similar electrochemical behavior to 3Na-PW$_{12}$ as shown in Supplementary Fig. 22a. Notably, the pH of the 4Na-SiW$_{12}$ electrolyte also changed reversibly in GCD process (Supplementary Fig. 22b), suggesting the same self-regulation mechanism as 3Na-PW$_{12}$. However, the 4Na-SiW$_{12}$ anolyte shows an initial discharge voltage of 1.25 V under the charge cut-off voltage of 1.8 V at 25 mA cm$^{-2}$, which is slightly lower than the 1.5 V output voltage of 3Na-PW$_{12}$-I$_2$ flow battery under the same discharging conditions. The results suggest that 3Na-PW$_{12}$ has better output performance than 4Na-SiW$_{12}$ as anolyte in the ARFBs, even though they exhibit similar electrochemical behavior.

In summary, we demonstrated a tungsten polyoxometalate based aqueous redox flow battery (3Na-PW$_{12}$-Br$_2$) with a high open-circuit voltage reaching up to 2.0 V. Sodium substituted tungsten polyoxometalates, such as 3Na-PW$_{12}$ and 4Na-SiW$_{12}$, were used as the anolyte, which has a low potential of −1.1 V and −1.19 V vs. SHE respectively. The self-regulation mechanism of 3Na-PW$_{12}$ anolyte for preventing water splitting was verified. The self-regulation mechanism of PW$_{12}$ was studied and confirmed by in-situ electrochemical measurements, [31]P NMR, Raman characterizations, DFT calculations and MD simulations. During the charging process, 3Na-PW$_{12}$ was

reduced and the pH value of the electrolyte was increased to 11 after each 3Na-PW$_{12}$ received 5 electrons. The [31]P NMR and titration analysis showed that 3Na-PW$_{12}$ was partially degraded into PW$_{11}$ and maintained the electrochemical activity, instead of fully decomposed into WO$_4{}^{2-}$. After the discharge, the electrolyte pH and PW$_{11}$ were restored to the initial state due to the self-healing property of polyoxometalates. Based on the self-regulation process of 3Na-PW$_{12}$, the flow battery offered a high open-circuit voltage of 2.0 V, capacity of 32 Ah l$^{-1}$, energy density of 36.5 Wh l$^{-1}$ and power density of 200 mW cm$^{-2}$ coupled with a Br$_2$/Br$^-$ catholyte. This study presents a promising anolyte candidate for the high voltage and high-power density ARFBs design.

## Methods
### Electrolyte preparation
The analytic pure (99%) H$_3$PW$_{12}$O$_{40}$ (H-PW$_{12}$) and H$_4$SiW$_{12}$O$_{40}$ (H-SiW$_{12}$) were purchased from Aladdin. The corresponding 0.1 mol l$^{-1}$ 1Na, 2Na and 3Na-PW$_{12}$ electrolyte was prepared by neutralization with 6 mol l$^{-1}$ NaOH (AR, MERYER). Deionized water is produced with water purifier (Qiqin, Q-LAB10-D1).

I$_2$/I$^-$ redox couple was primarily used in the flow battery due to its moderate redox potential (-0.54 V vs. SHE) and stable performance

across the tested pH range. $Br_2/Br^-$ can offer high redox potential (-1.07 V vs. SHE) to achieve high-voltage redox flow battery. Sodium iodide (NaI, 99%) and sodium bromide (NaBr, 99%) were purchased from Bidepharm. Both salts were dissolved in deionized water to prepare 1 mol l$^{-1}$ or 2 mol l$^{-1}$ solutions for different tests. The pH of the catholyte was adjusted with sulfuric acid to match the initial pH of the anolyte before the charging process.

## Electrochemical measurements

The Ag/AgCl electrode was calibrated in the electrochemical measurements. A platinum mesh electrode served as the reversible hydrogen electrode (RHE) in a sealed standard three-electrode system to correct the Ag/AgCl electrode (with a platinum plate counter electrode). The electrolyte was saturated KCl (pH 6.9) solution. Prior to the calibration, the electrolyte was continuously purged with hydrogen gas for at least 30 min to ensure a hydrogen-saturated state. Cyclic voltammetry was performed at the scan rate of 1 mV s$^{-1}$. The thermodynamic equilibrium potential for the hydrogen evolution reaction (HER) was determined by averaging the two interconversion points recorded in the hydrogen adsorption/desorption regions. All tests were conducted using an electrochemical workstation (Gamry INTERFACE 1010E), and the measured potentials of the Ag/AgCl electrode were calibrated to the RHE using the Nernst equation (Supplementary Fig. 23):

$$E_{HER} = -0.059 \times pH(V \text{ vs. SHE}) \tag{1}$$

$$E_{Ag/AgCl \text{ vs.SHE}} (\text{calibrated}) = E_{RHE} - E_{\text{measured RHE vs.Ag/AgCl}} \tag{2}$$

Three-electrode system was used to investigate the electrochemical properties of the prepared PW$_{12}$ electrolyte. Cyclic voltammetry (CV) measurements (without iR-correction) were conducted by an electrochemical working station (Gamry INTERFACE 1010E) and using GC electrode, Pt electrode and Ag/AgCl (saturated KCl) electrode worked as working electrode, counter electrode and reference electrode respectively. GC electrode and Ag/AgCl (saturated KCl solution) were used to measure the electrode potential under 1 nA current.

The Randles-Sevcik equation (the theoretical model for diffusion-controlled systems) was applied to the CV measurement row at different scan rates:

$$i_p = \left(2.69 \times 10^5\right) n^{2/3} A D^{1/2} C \sqrt{v} \tag{3}$$

where $i_p$ is peak current, $n$ is electron transfer number, $A$ is electrode area, $D$ is diffusion coefficient, $C$ is bulk concentration, $v$ is scan rate.

The diffusion coefficient of the electrolyte was measured by a rotating disk electrode (RDE, IPS, Elektroniklabor GmbH & Co. KG) with an o.d. 5 mm glass carbon disk. The measurements were carried out at a rotating speed from 50 rpm to 1200 rpm and a scan rate of 5 mV s$^{-1}$. The diffusion coefficient ($D$, cm$^2$ s$^{-1}$) is determined by the Levich equation:

$$i_{lim} = 0.62nAF\omega^{1/2}D^{2/3}v^{-1/6}C_0 \tag{4}$$

where $D$ is determined by the linear fitting of $i_{lim}$ (mA) against the square root of rotation rate ($\omega^{1/2}$ rad s$^{-1}$). Faraday constant ($F$) 96500 C mol$^{-1}$, electrode area ($A$) 0.196 cm$^2$, concentration ($C_O$) 0.01 mol l$^{-1}$, kinematic viscosity ($v$) cm$^2$ s$^{-1}$.

The kinetic current ($i_k$) was measured according to the Koutecky-Levich equation:

$$\frac{1}{i} = \frac{1}{i_k} + \frac{1}{i_{lim}} \tag{5}$$

where $i$ is measured current and $i_{lim}$ is limiting current. The exchange current $i_O$ is determined by Butler–Volmer equation (fitting the Tafel plot of log($i_k$) vs. overpotential ($\eta$)):

$$\eta = \frac{2.3RT}{\alpha F}\log(i_0) - \frac{2.3RT}{\alpha F}\log(i_k) \tag{6}$$

where the universal gas constant $R = 8.314$ J (K mol)$^{-1}$, $\alpha$ is the transfer coefficient, and $T$ is the temperature in kelvin. The standard rate constant $k^0$ was determined by the following equation:

$$i_0 = AFk^0C_0 \tag{7}$$

The $k^0$ is determined via the slope of the Tafel plot (Eqs. (4) and (5))

## Viscosity measurements

The kinematic viscosity (m$^2$ s$^{-1}$) of the electrolyte was measured by Ubbelohde viscometer based on the following equation:

$$v = kt \tag{8}$$

where $k$ is the kinematic viscosity constant $8.37 \times 10^{-8}$ m$^2$ s$^{-2}$ (25 °C), $t$ (s) is the time of liquid passing through two calibrated marks. The corresponding kinematic viscosity of the electrolyte was shown in Supplementary Table 4.

## Solubility measurements

The weight method was used to measure the solubility of 3Na-PW$_{12}$ in anolyte. An excessive amount of 3Na-PW$_{12}$ was added into 30 ml of water, then heated to dissolve it. The solution was maintained in a water bath at 25 or 35 °C with stirring for 24 h to obtain a saturated solution with 3Na-PW$_{12}$ precipitates. 10.00 ml of the clear saturated solution was quickly transferred into a beaker, and completely dried at 120 °C overnight. Weighed the dried 3Na-PW$_{12}$ solid, and it was noted as m. The saturated concentration of the 3Na-PW$_{12}$ solution at 25 or 35 °C was calculated by using the equation:

$$C_{Saturated} = m/0.01 \, (g \, l^{-1}) \tag{9}$$

## Computational methods

The systems were simulated by classical MD using the GROMACS 2024.0 with Amber 03 force field. Force field parameters for POMs and H$_3$O$^+$ were obtained by using Sobtop 1.0. Water was represented with the SPC model. The atom charge used in the simulations was the ChelpG charge. They were obtained with the Gaussian16 package at the DFT level (PBE0 functional) using the LANL2DZ basis set. Solvent effects were included in geometry optimizations by using the integral equation formalism polarizable continuum model (IEF-PCM) model implemented in Gaussian16. All simulations were performed with 3D-periodic boundary conditions using an atom cutoff of 10 Å for 1–4 van der Waals and 10 Å for 1-4 Coulombic interaction and corrected for long-range electrostatics by using the particle mesh Ewald (PME) summation method. The simulations were performed at 300 K starting with random velocities. The temperature was controlled by coupling the system to a thermal bath using the Velocity-rescaling algorithm with a relaxing time of 0.1 ps to keep the NVT canonical conditions throughout the simulation. Newton equations of motion were integrated using the leap-frog algorithm, and a time step of 1 fs. The systems were equilibrated with 5000 steps of energy minimization followed by simulations of 200 ps at NVT and 200 ps at NPT conditions. Finally, all systems were simulated for 10 ns under NVT conditions. In all cases, 50 POMs anions were embedded in a cubic solvent

box of 94³ Å³, as well as a number of $H_3O^+$ and $Na^+$ required to neutralize the charge of the system.

## Flow battery assembly and performance measurements

The battery was assembled with proton exchange membrane, gasket, graphite electrode (10 mm thick), current collector (Cu, 1.5 mm thick) and end plate (PP, 5 mm thick), as shown in Supplementary Fig. 24). Graphite felt electrode (2 mm thick) was used to increase the contact area between electrolyte and graphite electrode (Supplementary Fig. 25). The 'S' shape flow field with 2 mm depth and the area of 9 cm² and 16 cm² respectively were prepared. Nafion® 117 was used as the membrane in the experiment without pretreatment. 4 × 4 cm and 5 × 5 cm Nafion® 117 membranes were used for flow batteries with electrode areas of 9 cm² and 16 cm², respectively. The ohmic resistance of the flow battery (with electrode area of 16 cm²) was measured by electrochemical impedance spectroscopy (EIS) method by an electrochemical working station (Gamry INTERFACE 1010E). The measurements were performed in 3Na-PW₁₂(0.1 mol l⁻¹)-NaI (1 mol l⁻¹) and 3Na-PW₁₂(0.3 mol l⁻¹)-NaI (2 mol l⁻¹) electrolyte solution respectively, with the flow rate of 90 ml min⁻¹ at room temperature, in the frequency range of 0.01 Hz to 100 kHz and at the open-circuit voltage.

The performances of the flow battery including the galvanostatic charge/discharge (GCD) curves and cyclic tests were measured by a battery test system (LANHE CT3002N) with the flow rate of 90 ml min⁻¹. The polarization curves of the aqueous flow battery were measured using a continuous stepwise protocol by an electrochemical working station (Gamry INTERFACE 1010E). At different constant current densities, the potential was recorded for about 10 s to reach a stable condition. The state of charge (SOC) was not reset between different measurements and the battery underwent continuous discharge throughout the sweep. All the electrochemical performances of the aqueous flow battery were measured without iR-correction at room temperature.

To verify the reproducibility of the aqueous flow battery, a flow battery with the electrode area of 16 cm² was duplicated for the electrochemical performance measurements. The GCD curves and cyclic tests were measured under the same conditions described above by using a battery test system (LANHE CT3002N) with the electrolyte solution flow rate of 90 ml min⁻¹.

## In-situ CV, pH, and potential monitoring

The in-situ CV and electrode potential measurements were carried out using a modified device of flow cell, placed with a Ag/AgCl (saturated KCl) electrode worked as reference electrode and a GC electrode worked as working electrode within the channel of the cell (shown in Supplementary Fig. 26). The in-situ CV tests were carried out directly within the flow battery at the current density of 25 mA cm⁻² and scan rate of 100 mV s⁻¹ with 0.1 mol l⁻¹ 3Na-PW₁₂ worked as anolyte and 1 mol l⁻¹ NaI worked as catholyte. The flow battery stopped running for the CV measurement (about 30 s). After the CV measurement, the flow battery continued to charge or discharge for a specified time before the next CV test. The in-situ pH values of 0.1 mol l⁻¹ 3Na-PW₁₂ anolyte were obtained using a pH meter (INESA PHS-3C) during the charge-discharge process (25 mA cm⁻²) with 1 mol l⁻¹ NaI as catholyte (shown in Supplementary Fig. 27). The equilibrium potential for hydrogen evolution reaction (HER) was calculated using the Nernst equation with different pH values, as shown in Eq. (1).

## Electrolyte characterizations

During the charge-discharge, the reduced H-PW₁₂ anolyte was protected under $N_2$ atmosphere to prevent the oxidation by air. The PW₁₂ electrolyte solutions with different reduced states were quickly transferred into an NMR tube with the addition of $D_2O$ for ex-situ ³¹P NMR measurements. The ³¹P NMR characterizations were performed using a NMR spectrometer (Bruker Avance III 500 MHz). The Raman spectra of PW₁₂ and SiW₁₂ solutions before and after the charge and discharge experiments were recorded on a handheld Raman spectrometer (PERSERTECH, 785 nm). The laser power was 300 mW. UV-Vis absorption spectrum of the PW₁₂ and SiW₁₂ anolyte was measured on the UV-Vis spectrophotometer (SHIMADZU UV-2600i Series).

## Data availability

Data supporting the findings of this work are available within the paper and its Supplementary Information files and from the corresponding author upon reasonable request. The source data underlying Figs. 1–6 as well as all Supplementary Figs., and Supplementary Tables are available as a Source Data file. The atomic coordinates of the optimized computational models and the MD trajectories generated in this study have been deposited in the Figshare database under open access. (10.6084/m9.figshare.28868360). Source data are provided with this paper.

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

## Acknowledgements

The authors thank financial support from the National Natural Science Foundation of China (22278446), the science and technology innovation Program of Hunan Province (2024RC1010), and Hunan Provincial Natural Science Foundation of China (2022JJ10071).

## Author contributions

W.L. conceived and designed the project. W.P.L. mainly performed the experiments and analyzed experimental data. X.H. and L.T. helped to analyze part of the characterization. W.X. and W.S. helped with MD simulations. W.L., Y.D. and G.X. helped to write the paper. Z.S. carried out the DFT calculations. W.L., J.S., and B.Z. helped with experimental characterizations. All authors contributed to the discussion and revision of the paper.

## Competing interests

The authors declare no competing interests.
