## [Transparent Peer Review file · Nature Communications]

A Tungsten Polyoxometalate Mediated Aqueous Redox Flow Battery with High Open-Circuit Voltage up to 2 V

Corresponding Author: Professor Wei Liu

Version 0:

Reviewer comments:

Reviewer #1

(Remarks to the Author)

This is an interesting work which contributes significantly to the research area of flow batteries. The authors have addressed aqueous solutions of polyoxometallates (POM) as anolyte and performed thorough investigations combining experimental electrochemistry, NMR, Raman and UV-visible spectroscopy, as well as dft calculations and molecular dynamics (MD) simulations. The results obtained provide a new insight into the redox processes including POMs. The manuscript can be published in Nature Communications after a minor revision. My comments are listed below.

1. The authors should discuss briefly possible reasons of the POM degradation and the mechanistic role of solution pH in this process (see, for example a molecular modeling in the work by D.A. Tuzov et al. *Electrochim. Acta.* 2024 (20) 145182. <https://doi.org/10.1016/j.electacta.2024.145182>)

2. It was found that the ET rate constant for PW12 is larger by an order of magnitude as compared with that for V²⁺/V³⁺ redox couple. Some explanation should be given to elucidate this finding based on the Marcus theory.

3. It is not clear why the authors used Mulliken atomic charges in the force field when performing MD simulations. It would be more reasonable to deal with the ChelpG charges.

Reviewer #2

(Remarks to the Author)

This work presents new chemistries for the anolyte of aqueous redox flow batteries, based on the sodium salts of the Keggin-type plenary anion (PW12 and SiW12). Polyoxometalates have already shown their potential as electroactive species and within this work, the possibility to obtain a low potential for the anolyte is shown. This research is of great relevance for the scientific community, as it is framed in the current energy transition, one of the main concerns of the global population nowadays.

The work is well organized, and all the conclusions are supported by enough experimental data. Although I consider the paper deserves publication in Nature Communications, some minor revisions should be addressed before publication.

In my opinion, the quality of the work is high in terms of methodology and results, but the article lacks clarity. Even though the introduction properly contextualizes the work in the current state of the art and its importance, I would suggest including a more detailed explanation of the main objective of the work in the introduction, or an introductory paragraph before the results for a better understanding of the work. For example, I consider that the comparison between the performance of the PW12 and the SiW12 is relevant for the work, but SiW12 is not mentioned until line 315. It happens the same with the utilization of two catholytes (Br and I), it is confusing.

In addition, some experimental details should be included, either in the experimental section or in the supporting information:

- Which kind of graphite electrode is employed? Is it compressed?
- Concentration and details of how the catholyte is prepared or purchased. In addition, include an explanation of why Br/Br⁻ and I/I⁻ were tested and when each was used.
- How was the data held in Fig.1.b obtained? taking aliquots from the cell? During ex-situ charge-discharge?
- Include calculations of the theoretical HER potential.

- How were the reduced forms of the H-PW12 for the structural and electroactive tests obtained? During the charge-discharge? Ex-situ? How were preserved to avoid re-oxidation? Explain the procedure.

Although the English is understandable, minor mistakes in the spelling or grammar must be corrected:

- In-situ
- Line 52: Dawson structure
- Line 68: we present
- Fig.2.e(iv): Theoretical
- Line 138: substitution

Reviewer #3

(Remarks to the Author)

The authors introduce a novel, low-potential anolyte active material in the form of Na-substituted phosphotungstic acid that provides full-cell voltages of aqueous RFBs of up to 2.0 V. The submitted manuscript addresses a critical challenge in the field of aqueous flow batteries by increasing the voltage output beyond the water-splitting potentials. I agree with the authors that this work represents noteworthy results within the flow battery community and therefore recommend a publication in Nature Communications.

Overall, the English used in this manuscript is mostly of high standard. The graphics are of good quality and provide all the necessary information. Additionally, the research presented in this work is of high quality.

However, some sections need improvements in terms of their structure. Some graphics are not sufficient or are not described at all within the text. Furthermore, some graphics need slight enhancements. Due to the statement above, I suggest a publication after a major revision following the detailed comments below.

Detailed comments:

1) Minor grammar mistakes were apparent that need to be corrected:

- Page 3, Line 24: Do the authors mean “energy resources” instead of “energy”?
- Page 3, Line 32: Either change to “The all-vanadium battery” or the plural “All-vanadium batteries”.
- Page 5, Line 79: Please change to “batteries”.
- Page 7, Line 94: Change to “which leads to the increase of the pH of the anolyte.”
- Page 17, Line 270: Change “detailly investigated” to “investigated in detail”.
- Page 18, Line 293: Please change “oxygen oxidation” to “oxidation by atmospheric oxygen”.
- Page 19, Line 316: Please change “UV-visible measurements” to “UV-Vis spectroscopy measurements”.
- Page 20, Line 344: Please change to “with a Br₂/Br⁻ catholyte”.

2) Page 3, Line 37: I recommend using the term “active material” instead to be more precise.

3) Figure 1:

- Please add to the description of Figure 1 b a specific description of the applied electrolyte, including the active material concentration as well as the supporting electrolyte.

4) Page 7, Line 95-97: At this point, you did not yet show any results that prove those statements. Please refer to the corresponding experiments / Figures that support your statements.

5) Page 7, Line 97: Please refer to the section within the manuscript or supporting information that describes the corresponding experiment to Figure 1b.

6) Page 8, Figure 2:

- Add the electrolyte concentration within the used Ag/AgCl reference electrode
- From the details about your measurement cell provided in the supporting information, it seems that the in-situ CVs were measured directly within the flow cell. Was the flow and the cyclization stopped during the measurement of each CV? Please provide more information on how the measurements were conducted.

7) Page 9, Line 138: Do the authors mean “substitution” instead of “substation”?

8) Page 10, Line 156: “were cause by the charge excitation of Od→W and Ob, Oc→W respectively”. Please add a citation that proves this statement or rephrase it to an assumption instead of a general statement.

9) Page 14, Line 233: Please briefly describe what DFT and MD methods were used and refer to the methods section for the reader to find more details on the methods.

10) Figure 5:

- Please ensure that all variables (e.g. i_k and k°) are written in italics that can be found within the graphic.
- The color gradient in figure 5a, 5b and 5d is hardly visible. Also, ensure that only Figures that depend on each other share the same color gradient to avoid ambiguity. Figure 5a shows CVs, while 5d shows the Koutecky-Levich plot corresponding to Figure 5b but shares the same color gradient as 5a. Please, rework the color gradients of the whole Figure. Add the measurement points that were used for the Levich as well as Koutecky-Levich plot in Figure 5b. If possible, also include a color gradient of each data point in Figures 5c and 5e, referring to Figures 5b and 5d, respectively.

11) Page 17, Line 271: “Which implies...” – Please briefly describe how the CVs behave with the change of the scan rates / provide the reader more details on what exactly implies the diffusion limitation.

12) Figure 6:

- If possible, add an additional y-axis showing the theoretical equivalence of transferred electrons per capacity or mark the endpoint / maximum reachable capacity with the corresponding electrons taking the theoretical capacity of the cells into account.
- The text underneath is missing a description and evaluation of Figure 6f. Please add a short corresponding paragraph.
- On page 18, Line 302, Figure 6c is roughly described but never referred to. Please always introduce the Figure first, then describe the plot, and afterwards interpret the Figure to provide more clarity to the reader. Please check the whole manuscript after this scheme and rephrase the necessary paragraphs.

- Figure 6c: Please provide or refer to a description of the corresponding experiment to measure the polarization curves. How was each measurement point taken? What was the SOC of the battery, and was it reset for each measurement point, or was it measured in a continuous sweep?
- 13) Page 19, Line 316: Please describe the UV-Vis spectra. How do they indicate that replacing the protons has no change in the structure?
- 14) Page 19, Line 318-319: Please describe more precisely which redox-potential / corresponding electron transfer reaction is here referred to.
- 15) Page 20, Line 336: Please add a short list of experiments that have proven this mechanism within this work.
- 16) All in all, the manuscript provides extensive electrochemical characterizations of the 3Na-PW12 species. However, another vital parameter for future applications is the solubility of the active material. I highly recommend the authors add a statement on the maximum solubility of the material. If necessary, please provide additional experiments.
- 17) Some critical information is missing in the methods section:
 - The “Electrolyte preparation” is missing the purity in % of the precursors as well as all further used chemicals.
 - “Electrochemical measurements” is missing the information on which potentiostat and RDE were applied. Since measurement errors depend on the devices, this information is necessary to provide reproducibility. Furthermore, the surface area / diameter of the applied RDE tip and the electrode material (presumable glassy carbon) are missing.
 - Please provide the names of the Raman and UV-Vis spectrometer that was used.

Besides, smaller typos were apparent:

- Page 3, Line 35: In “1.5V” a space is missing between number and unit.
- Page 17, Line 274: Do the authors mean “RDE” instead of “RED”?
- Please make sure to use always protected spaces and dashes to avoid line breaks within units or between numbers and units, e.g.
 - o Page 12, Line 195: of HxPO4(3-x)-
 - o Page 16, Line 264: 0.1 mol l-1
 - o Page 20, Line 342: 2.0 V
 - o Page 20, Line 343: 200 mW cm-1

Version 1:

Reviewer comments:

Reviewer #1

(Remarks to the Author)

The authors have properly addressed my comments and the revised manuscript can be published as is.

Reviewer #2

(Remarks to the Author)

I think the paper is now ready for publication.

Reviewer #3

(Remarks to the Author)

The authors introduce a novel, low-potential anolyte active material in the form of Na-substituted phosphotungstic acid that provides full-cell voltages of aqueous RFBs of up to an OCV of 2.0 V.

The revised manuscript addresses all reviewer suggestions in depth. The authors took sufficient time to consider all the comments and adjusted the manuscript accordingly. However, during the revision process, more minor typos became evident, and I would like to request some additional information:

- Page 20, Line 344: I thank the authors for providing the requested information. However, could you please briefly add the method used to obtain those results? Did you use gravimetric or UV-Vis measurements? If possible, also add the corresponding data to the supporting information.
- Page 20, Line 349: Do the authors mean “spectra” instead of “spectroscopy”?
- Page 23, Line 406: I would suggest a rephrasing of “was calculated in the CV measurements with different scan rates” to “was applied to the CV measurement row at different scan rates”.
- Page 27, Line 476: Do the authors mean “worked” instead of “worded”?

Response to reviewers:

Reviewer #1 (Remarks to the Author):

This is an interesting work which contributes significantly to the research area of flow batteries. The authors have addressed aqueous solutions of polyoxometallates (POM) as anolyte and performed thorough investigations combining experimental electrochemistry, NMR, Raman and UV-visible spectroscopy, as well as dft calculations and molecular dynamics (MD) simulations. The results obtained provide a new insight into the redox processes including POMs. The manuscript can be published in Nature Communications after a minor revision. My comments are listed below.

Reply: Thank you for your positive comments and professional suggestions.

Q1. The authors should discuss briefly possible reasons of the POM degradation and the mechanistic role of solution pH in this process (see, for example a molecular modeling in the work by D.A. Tuzov et al. *Electrochim. Acta.* 2024 (20) 145182. <https://doi.org/10.1016/j.electacta.2024.145182>)

Reply: Thank you for your question. Polyoxometalates (POMs) are a large group of anionic polynuclear metal–oxo clusters and the Keggin anion, $\text{PW}_{12}\text{O}_{40}^{3-}$ (PW_{12}), is one of the most representative polyoxometalates (POMs)^{1,2}. As described in the recommended reference, polyoxometalate is commonly stable in mildly acidic and neutral aqueous solutions, but the degradation becomes remarkably feasible in alkali media³. The reason is H^+ ions interact with the surface oxygen atoms of the PW_{12} in acidic solutions, which reduces the negative charge density of polyoxometalate framework and effectively suppresses the repulsion-driven structural dissociation^{4,5}. However, due to the coordination effect of OH^- with metal ions, the alkaline medium leads to the weakening of W-O-W bonds of PW_{12} and subsequent structural decomposition of polyoxometalate frameworks². This process can be expressed as the following equation:

With continuous increase of pH, $\text{PW}_9\text{O}_{34}^{9-}$ could be further formed:

Related discussion was added into the manuscript Page 9 and 10, line153-158. The reaction equations of the PW_{12} decompositions were added to the supplementary Figure 5.

Q2. It was found that the ET rate constant for PW_{12} is larger by an order of magnitude as compared with that for $\text{V}^{2+}/\text{V}^{3+}$ redox couple. Some explanation should be given to elucidate this finding based on the Marcus theory.

Reply: Thank you for your question. According to the Marcus equation (1) and Eyring

Equation (2):

$$\Delta G^\ddagger = \frac{\lambda}{4} \left(1 + \frac{\Delta G^0}{\lambda}\right)^2 \quad (1)$$

$$k = \frac{k_B T}{h} e^{\left(-\frac{\Delta G^\ddagger}{RT}\right)} \quad (2)$$

$$k \propto -\frac{(\lambda + \Delta G^0)^2}{4\lambda RT} \quad (3)$$

where ΔG^\ddagger is activation energy, ΔG^0 is Gibbs energy change, λ is reorganization energy, k is the reaction rate constant, k_B is Boltzmann's constant, and h is Planck's constant. Compared to V^{2+}/V^{3+} , PW_{12} exhibits lower reorganization energy (λ) due to its structural rigidity (suppressing geometric distortion) and large size (reducing solvent reorganization)^{3,6}. At the same time, the electron-delocalization of polyoxometalate framework facilitates efficient electron transfer pathways, resulting in higher reaction rates, as claimed in the reported literatures⁷.

Related discussion was added into the manuscript, Page 18, line 301-305, yellow highlighted.

Q3. It is not clear why the authors used Mulliken atomic charges in the force field when performing MD simulations. It would be more reasonable to deal with the ChelpG charges.

Reply: Thanks for your professional suggestions in MD simulation. According to the suggestion, the supplementary MD simulations were conducted based on ChelpG charge by Gromacs. The DFT calculations were performed with the Gaussian16 package at the DFT level (PBE0 functional) using the LANL2DZ basis set. Solvent effects were included in geometry optimizations by using the integral equation formalism polarizable continuum model (IEF-PCM) implemented in Gaussian16.

Figure R1 shows the RDFs curves of PW_{12} at different reduced states. The $4e^-$ reduced PW_{12} ($4e^-$ - PW_{12}) tends to attract more H_3O^+ and Na^+ in solution than $0e^-$ - PW_{12} , which explains the pH change of $3Na$ - PW_{12} anolyte during the charge-discharge process. The distance between $4e^-$ - PW_{12} anions is longer than that of $0e^-$ - PW_{12} anions (Supplementary Fig. 9c), because of the strong interaction between cations (H_3O^+ or Na^+) and $4e^-$ - PW_{12} (Supplementary Fig. 9d).

Related discussion was added into the manuscript, Page 15, line 246-250 and Page 16, line 271-273, yellow highlighted. Figure R1a was added into the manuscript (Figure 4d), and Figure R1b was added into the Supplementary Information (Supplementary Fig. 10).

Figure R1| a. POM-H₃O⁺ Radial Distribution Functions (RDFs) calculated from classical MD simulations using the center of mass of each PW₁₂ as reference. b. POM-Na⁺ Radial Distribution Functions (RDFs) calculated from classical MD simulations using the center of mass of each PW₁₂ as reference.

Reviewer #2 (Remarks to the Author):

This work presents new chemistries for the anolyte of aqueous redox flow batteries, based on the sodium salts of the Keggin-type plenary anion (PW₁₂ and SiW₁₂). Polyoxometalates have already shown their potential as electroactive species and within this work, the possibility to obtain a low potential for the anolyte is shown. This research is of great relevance for the scientific community, as it is framed in the current energy transition, one of the main concerns of the global population nowadays. The work is well organized, and all the conclusions are supported by enough experimental data. Although I consider the paper deserves publication in Nature Communications, some minor revisions should be addressed before publication.

Reply: Thank you for your positive comments.

Major points:

Q4. In my opinion, the quality of the work is high in terms of methodology and results, but the article lacks clarity. Even though the introduction properly contextualizes the work in the current state of the art and its importance, I would suggest including a more detailed explanation of the main objective of the work in the introduction, or an introductory paragraph before the results for a better understanding of the work. For example, I consider that the comparison between the performance of the PW₁₂ and the SiW₁₂ is relevant for the work, but SiW₁₂ is not mentioned until line 315. It happens the same with the utilization of two catholytes (Br and I), it is confusing.

Reply: Thank you for your constructive feedback.

Both the PW₁₂ and SiW₁₂ are typical kegging structured polyoxometalates (POMs) and widely studied in electrochemistry and catalysis. They have similar structure and

electrochemical properties, making them ideal for exploring the self-regulation performance in this study. To clarify the objective of this work, we have rewritten the last paragraph of the Introduction in manuscript. Please see the manuscript page 5, line 69-83, yellow highlighted.

Halogen catholytes were used in this aqueous flow battery. The I⁻/I couple was used primarily due to its moderate redox potential (~0.54 V vs. SHE) and stable performance across the tested pH range. The Br⁻/Br couple with a higher redox potential (~1.07 V vs. SHE) than I⁻/I couple was selectively tested in flow battery experiments to evaluate its potential for achieving higher cell voltages. We also added a brief description in the manuscript Page 5, line 70 and Page 19, line 317-319, yellow highlighted. We hope these changes may improve clarity of our manuscript. We appreciate your valuable suggestions.

Q5. In addition, some experimental details should be included, either in the experimental section or in the supporting information.

Reply: Thank you for your suggestion. We have added experimental details about electrolyte preparation, in-situ measurements, and electrolyte characterizations in the experimental section, as shown in the manuscript, page 22-27, yellow highlighted.

Q6. Which kind of graphite electrode is employed? Is it compressed?

Reply: Thank you for your question. Graphite felt was employed in this polyoxometalate mediated flow battery. As shown in Supplementary Fig. 21, the graphite felt is woven by graphite fibers with a diameter of 20 μm. Therefore, the electrode has a large surface area for the electrolyte discharge. The photographs of the employed graphite felt electrode were shown below and were added in Supplementary Figure 21. The related description was added into the manuscript, Page 26 line 458-459, yellow highlighted.

Supplementary Fig. 21| The digital photo of the employed commercial graphite felt electrode in flow battery: (a) The front view, (b) The side view. Optical micrographs of graphite felts (c) magnification of 100x, (d) magnification of 400x.

Q7. Concentration and details of how the catholyte is prepared or purchased. In addition, include an explanation of why Br^-/Br^- and I^-/I^- were tested and when each was used.

Reply: Thank you for your question. We have added the catholyte preparation method in the manuscript, the experimental section, Page 23, line 392-398, yellow highlighted.

Catholyte Preparation:

Sodium iodide (NaI , 99%) and sodium bromide (NaBr , 99%) were purchased from Bidepharm. Both salts were dissolved in deionized water to prepare 1 mol l^{-1} or 2 mol l^{-1} solutions for different tests. The pH of the catholyte was adjusted with sulfuric acid to match the initial pH of the anolyte before the charging of the battery.

The I^-/I^- couple was used primarily due to its moderate redox potential ($\sim 0.54 \text{ V}$ vs. SHE) and stable performance across the tested pH range. The Br^-/Br^- couple with a higher redox potential ($\sim 1.07 \text{ V}$ vs. SHE) than I^-/I^- couple was selectively tested in flow battery experiments to evaluate its potential for achieving higher cell voltages. Therefore, I_2/I^- electrolyte was mainly used in the in-situ electrochemical tests (Figure 2), in-situ pH tests (Figure 2) and capacity performance measurements of the flow battery (Figure 6d-f). Br_2/Br^- electrolyte was mainly used in the polarization curves and capacity performance measurements of the flow battery (Figure 6b-c).

We have added the specific catholyte in the caption of Figure 6b, yellow highlighted. The reason for using Br⁻/Br couple in the measurement was described in the manuscript page 19, line 317-318, yellow highlighted.

Q8. How was the data held in Fig.1.b obtained? taking aliquots from the cell? During ex-situ charge-discharge?

Reply: Thank you for your question. The pH values of 0.1 mol l⁻¹ 3Na-PW₁₂ anolyte were obtained via in-situ pH test during the charge-discharge process (25 mA cm⁻²) with 1 mol l⁻¹ NaI as catholyte. The device of the in-situ pH test is shown below and added in the Supplementary Figure 23. The related experimental details were added in the revised manuscript, page 27, line 480, and the caption of Figure 1b, yellow highlighted.

Supplementary Figure 23| The diagram of in-situ pH test of 0.1 mol l⁻¹ 3Na-PW₁₂ anolyte during charge-discharge process.

Q9. Include calculations of the theoretical HER potential.

Reply: Thanks for your question. We have added the theoretical HER potential calculations to the revised manuscript, page 27, line 480-483. The equilibrium potential for HER was calculated using the Nernst equation with different pH values:

$$E_{\text{HER}} = -0.059 \times \text{pH} \text{ (V vs. SHE)}$$

Q10. How were the reduced forms of the H-PW₁₂ for the structural and electroactive tests obtained? During the charge-discharge? Ex-situ? How were preserved to avoid re-oxidation? Explain the procedure.

Reply: Thanks for your comments. During the charge-discharge, the reduced H-PW₁₂ anolyte was protected under N₂ atmosphere to prevent the oxidation by air. The PW₁₂ electrolyte solutions with different reduced states were quickly transferred into an NMR tube with the addition of D₂O for ex-situ ³¹P NMR measurements. The related description was added in the experimental section, Page 27, line 485-490.

The electrochemical tests, such as CV curve and redox potential measurements, were

carried out in-situ within the flow battery, by using the electrochemical test device as shown below (Supplementary Fig. 22). The related description was added in the experimental section, Page 26, line 469-482.

Supplementary Fig. 22| In-situ electrochemical test device.

Q11. Although the English is understandable, minor mistakes in the spelling or grammar must be corrected:

- In-situ
- Line 52: Dawson structure
- Line 68: we present
- Fig.2.e(iv): Theoretical
- Line 138: substitution

Reply: Thanks a lot for your reminder, the mistakes mentioned have been corrected.

Reviewer #3 (Remarks to the Author):

The authors introduce a novel, low-potential anolyte active material in the form of Na-substituted phosphotungstic acid that provides full-cell voltages of aqueous RFBs of up to 2.0 V. The submitted manuscript addresses a critical challenge in the field of aqueous flow batteries by increasing the voltage output beyond the water-splitting potentials. I agree with the authors that this work represents noteworthy results within the flow battery community and therefore recommend a publication in Nature Communications. Overall, the English used in this manuscript is mostly of high standard. The graphics are of good quality and provide all the necessary information. Additionally, the research presented in this work is of high quality.

However, some sections need improvements in terms of their structure. Some graphics are not sufficient or are not described at all within the text. Furthermore, some graphics need slight enhancements. Due to the statement above, I suggest a publication after a major revision following the detailed comments below.

Reply: Thanks for your positive comments and professional suggestions to improve this work.

Q12. Minor grammar mistakes were apparent that need to be corrected:

- Page 3, Line 24: Do the authors mean “energy resources” instead of “energy” ?

- Page 3, Line 32: Either change to “The all-vanadium battery” or the plural “All-vanadium batteries” .
- Page 5, Line 79: Please change to “batteries” .
- Page 7, Line 94: Change to “which leads to the increase of the pH of the anolyte.”
- Page 17, Line 270: Change “detailedly investigated” to “investigated in detail” .
- Page 18, Line 293: Please change “oxygen oxidation” to “oxidation by atmospheric oxygen” .
- Page 19, Line 316: Please change “UV-visible measurements” to “UV-Vis spectroscopy measurements” .
- Page 20, Line 344: Please change to “with a Br₂/Br⁻ catholyte” .

Reply: Thank you very much for your help. The grammar mistakes mentioned above were corrected.

Q13. Page 3, Line 37: I recommend using the term “active material” instead to be more precise.

Reply: Thank you for your comments. The term mentioned above was replaced with “active material”.

Q14. Figure 1:

- Please add to the description of Figure 1 b a specific description of the applied electrolyte, including the active material concentration as well as the supporting electrolyte.

Reply: Thanks for your suggestion. The detailed information of Figure 1b was added in the figure caption: The cyclic pH change of 3Na-PW₁₂ anolyte (0.1 mol l⁻¹), obtained by in-situ pH monitor during a complete charge and discharge process with 1 mol l⁻¹ NaI as catholyte.

Q15. Page 7, Line 95-97: At this point, you did not yet show any results that prove those statements. Please refer to the corresponding experiments / Figures that support your statements.

Reply: Thanks for your suggestion. The corresponding statements were changed as follows: “After fully charging, the captured protons could be gradually released back into the anolyte during the discharging process, verified by Fig. 1b.”, as shown in Page 7, Line 102, yellow highlighted.

Q16. Page 7, Line 97: Please refer to the section within the manuscript or supporting information that describes the corresponding experiment to Figure 1b.

Reply: Thanks for your suggestion. The corresponding experimental details were added in the revised manuscript, Page 27, line 478. The in-situ pH values of 0.1 mol l⁻¹ 3Na-

PW₁₂ anolyte were obtained using a pH meter (INESA PHS-3C) during the charge-discharge process (25 mA cm⁻²) with 1 mol l⁻¹ NaI as catholyte (shown in Supplementary Fig. 23). Also, the caption of Figure 1b was modified as: The cyclic pH change of 3Na-PW₁₂ anolyte (0.1 mol l⁻¹), which obtained by in-situ pH monitor during a complete charge and discharge process with 1 mol l⁻¹ NaI as catholyte.

Q17. Page 8, Figure 2:

- Add the electrolyte concentration within the used Ag/AgCl reference electrode

Reply: Thanks for your suggestion. The Ag/AgCl reference electrode with saturated KCl solution as electrolyte was used as reference electrode in this work. The information was added in the manuscript Page 23, line 402-404.

- From the details about your measurement cell provided in the supporting information, it seems that the in-situ CVs were measured directly within the flow cell. Was the flow and the cyclization stopped during the measurement of each CV? Please provide more information on how the measurements were conducted.

Reply: Thanks for your comments. The in-situ CVs curves were measured directly within the flow cell. The in-situ CV tests were carried out at the current density of 25 mA cm⁻² and scan rate of 100 mV s⁻¹ with 0.1 mol l⁻¹ 3Na-PW₁₂ worked as anolyte and 1 mol l⁻¹ NaI worked as catholyte. The flow battery stopped running for the CV measurement (about 30 seconds). After the CV measurement, the flow battery continued to charge or discharge for a specified time before the next CV test. This information was also updated in the revised manuscript, Page 27, line 473.

Q18. Page 9, Line 138: Do the authors mean “substitution” instead of “substation” ?

Reply: Thanks for your suggestion and the typo was corrected.

Q19. Page 10, Line 156: “were cause by the charge excitation of O_d→W and O_b, O_c →W respectively” . Please add a citation that proves this statement or rephrase it to an assumption instead of a general statement.

Reply: Thanks for your suggestion. The corresponding reference was cited as number 28 in Page 10, line 168.

28. Himeno, S., Takamoto, M. & Ueda, T. Synthesis, characterisation and voltammetric study of a b-Keggin-type [PW₁₂O₄₀]³⁻ complex. *Journal of Electroanalytical Chemistry* (1999) doi:10.1016/S0022-0728(99)00068-6.

Q20. Page 14, Line 233: Please briefly describe what DFT and MD methods were used and refer to the methods section for the reader to find more details on the methods.

Reply: Thanks for the suggestions. A brief introduction of DFT and MD methods was added in the corresponding paragraph, Page 15, line 246-250, yellow highlighted.

Q21. Figure 5:

- Please ensure that all variables (e.g. i_k and k°) are written in italics that can be found within the graphic.
- The color gradient in figure 5a, 5b and 5d is hardly visible. Also, ensure that only Figures that depend on each other share the same color gradient to avoid ambiguity. Figure 5a shows CVs, while 5d shows the Koutecky-Levich plot corresponding to Figure 5b but shares the same color gradient as 5a. Please, rework the color gradients of the whole Figure. Add the measurement points that were used for the Levich as well as Koutecky-Levich plot in Figure 5b. If possible, also include a color gradient of each data point in Figures 5c and 5e, referring to Figures 5b and 5d, respectively.

Reply: Thanks for the suggestions. We changed the color in figure 5 to avoid ambiguity. The variables within the graphs were modified as italics. The Figures 5a, 5b and 5d were redrawn, as shown in the revised manuscript, Page 17.

Q22. Page 17, Line 271: “Which implies…” – Please briefly describe how the CVs behave with the change of the scan rates / provide the reader more details on what exactly implies the diffusion limitation.

Reply: Thanks for your question. According to the Randles-Sevcik equation, which is the theoretical model for diffusion-controlled systems:

$$i_p = (2.69 \times 10^5) n^{2/3} A D^{1/2} C \sqrt{v}$$

where i_p is peak current, n is electron transfer number, A is electrode area, D is diffusion coefficient, C is bulk concentration, v is scan rate.

The peak currents (i_p) were linearly fitted with the square root of the scan rate (v) (Supplementary Fig. 11), which implies that the charge and discharge reactions of 3Na-PW₁₂ are dominated by diffusion kinetics.

Related discussion was added in the revised manuscript, Page 17, line 287-290.

Q23. Figure 6:

- If possible, add an additional y-axis showing the theoretical equivalence of transferred electrons per capacity or mark the endpoint / maximum reachable capacity with the corresponding electrons taking the theoretical capacity of the cells into account.

Reply: Thanks for the suggestion, we have modified the corresponding Figure 6b on Page 18.

- The text underneath is missing a description and evaluation of Figure 6f. Please add a short corresponding paragraph.

Reply: The description of Figure 6f was added. Figure 6f shows the cycle performances of the assembled 3Na-PW₁₂-NaI flow battery, which showed stable operations for 40 cycles.

- On page 18, Line 302, Figure 6c is roughly described but never referred to. Please always introduce the Figure first, then describe the plot, and afterwards interpret the Figure to provide more clarity to the reader. Please check the whole manuscript after this scheme and rephrase the necessary paragraphs.

Reply: Thanks for your suggestion. We have added the description about Figure 6c in the revised manuscript, Page 20, line 340. Also, we have checked the whole manuscript and related paragraphs were modified.

- Figure 6c: Please provide or refer to a description of the corresponding experiment to measure the polarization curves. How was each measurement point taken? What was the SOC of the battery, and was it reset for each measurement point, or was it measured in a continuous sweep?

Reply: Thanks for your comments. The polarization curves in Figure 6c were measured using a continuous stepwise protocol. At different constant current densities, the potential was recorded for about 10 seconds to reach a stable condition. The state of charge (SOC) was not reset between different measurements and the battery underwent continuous discharge throughout the sweep. The experimental details were added in the experimental section on Page 26, line 464-468, yellow highlighted.

Q24. Page 19, Line 316: Please describe the UV-Vis spectra. How do they indicate that replacing the protons has no change in the structure?

Reply: Thanks for your suggestion. UV-Vis spectroscopy measurements show similar absorption peaks at 206 and 260 nm with different Na substitutions of SiW₁₂, which could be ascribed to charge excitation of W-O_d and W-O_{b,c} bonds respectively. These results indicate that replacing the protons with 4 sodium ions has no significant structural changes to the SiW₁₂ frameworks (Supplementary Fig. 17).

Related discussion was added in Page 21 line 348-354, yellow highlighted.

Q25. Page 19, Line 318-319: Please describe more precisely which redox-potential / corresponding electron transfer reaction is here referred to.

Reply: Thanks for the question. The CV curve of 4-Na SiW₁₂ shows 4 characteristic peaks, which can be ascribed to 4 single electron transfer reactions under the neutral concentration, as follows:⁸

Related discussion was added in Page 21, line 353, yellow highlighted. Related equations were added in the caption of Supplementary Fig. S18.

Q26. Page 20, Line 336: Please add a short list of experiments that have proven this mechanism within this work.

Reply: Thanks for the suggestion and we have added a short list of experiments for the mechanism study.

The self-regulation mechanism of PW_{12} was studied and confirmed with:

- In-situ measurements including pH monitoring, redox potential test of the anolyte, CV measurements
- P-NMR
- Raman
- DFT calculation
- MD simulations

The related discussion was added in the conclusion part, Page 22, line 373-375, yellow highlighted.

Q27. All in all, the manuscript provides extensive electrochemical characterizations of the 3Na-PW_{12} species. However, another vital parameter for future applications is the solubility of the active material. I highly recommend the authors add a statement on the maximum solubility of the material. If necessary, please provide additional experiments.

Reply: Thank you for your suggestion. The solubility of the 3Na-PW_{12} anolyte was measured to be 0.38 mol l^{-1} (1119 g l^{-1}) at $25 \text{ }^\circ\text{C}$ and 0.45 mol l^{-1} (1325 g l^{-1}) at $25 \text{ }^\circ\text{C}$, which means that the theoretical capacities of 50.9 Ah l^{-1} and 60.3 Ah l^{-1} , respectively, of the anolyte could be reached (shown in Supplementary Fig. 16).

The results were described in the revised manuscript, Page 20, line 343-347.

Supplementary Fig. 16| The solubility of 3Na-PW₁₂ at 25 and 35 °C.

Q28. Some critical information is missing in the methods section:

- The “Electrolyte preparation” is missing the purity in % of the precursors as well as all further used chemicals.
- “Electrochemical measurements” is missing the information on which potentiostat and RDE were applied. Since measurement errors depend on the devices, this information is necessary to provide reproducibility. Furthermore, the surface area / diameter of the applied RDE tip and the electrode material (presumably glassy carbon) are missing.
- Please provide the names of the Raman and UV-Vis spectrometer that was used.

Reply: Thanks for your questions.

(1) The applied chemicals were shown in Table R1 below. Related information was added in the **experimental section** in the revised manuscript on Page 22-23, yellow highlighted.

Table R1. The applied chemicals in this work

Name	Purity	Brand
H ₃ PW ₁₂ O ₄₀	AR	Aladdin
H ₄ SiW ₁₂ O ₄₀	AR	Aladdin
NaI	99%	Bidepharm
NaBr	99%	Bidepharm
H ₂ SO ₄ (98%)	AR	KESHI
NaOH	≥98%	MERYER

(2) The PW₁₂ electrolyte solutions with different reduced states were quickly transferred into an NMR tube with the addition of D₂O for ³¹P NMR measurements. The ³¹P NMR characterizations were performed using a NMR spectrometer (Bruker Avance III 500 MHz). The Raman spectra of PW₁₂ and SiW₁₂ solutions before and after the charge and discharge experiments were recorded on a handheld Raman spectrometer (PERSERTECH, 785 nm). The laser power was 300 mW. UV-Vis absorption spectrum of the PW₁₂ and SiW₁₂ anolyte was measured on the UV-Vis spectrophotometer (SHIMADZU UV-2600i Series). The performances of the flow battery including the galvanostatic charge/discharge (GCD) curves and cycling tests were measured by a battery test system (LANHE CT3002N). The diffusion coefficient of the electrolyte was measured by a rotating disk electrode (with an o.d. 5 mm glass carbon disk, IPS RDE 201 111-B). Cyclic voltammetry (CV) was conducted by an electrochemical working station (Gamry INTERFACE 1010E) and using GC electrode, Pt electrode and Ag/AgCl (saturated KCl) electrode as working electrode, counter electrode and reference electrode respectively.

The experimental details were added in the experimental part, Page 22-28, yellow highlighted.

Q29.

Besides, smaller typos were apparent:

- Page 3, Line 35: In “1.5V” a space is missing between number and unit.
- Page 17, Line 274: Do the authors mean “RDE” instead of “RED” ?
- Please make sure to use always protected spaces and dashes to avoid line breaks within units or between numbers and units, e.g.
 - o Page 12, Line 195: of H_xPO₄^{(3-x)-}
 - o Page 16, Line 264: 0.1 mol l⁻¹
 - o Page 20, Line 342: 2.0 V
 - o Page 20, Line 343: 200 mW cm⁻¹

Reply: The mistakes mentioned above have been corrected. Thank you very much for your help to improve this manuscript.

Reference

1. I. Gumerova, N. & Rompel, A. Polyoxometalates in solution: Speciation under spotlight. *Chemical Society Reviews* **49**, 7568–7601 (2020).
2. López, X., Nieto-Draghi, C., Bo, C., Avalos, J. B. & Poblet, J. M. Polyoxometalates in Solution: Molecular Dynamics Simulations on the α -PW₁₂O₄₀³⁻ Keggin Anion in

- Aqueous Media. *J. Phys. Chem. A* **109**, 1216–1222 (2005).
3. Tuzov, D. A., Zueva, E. M., Zinkicheva, T. T. & Nazmutdinov, R. R. Co⁴⁺-polyoxometalates in aqueous solutions. Stability, ion pairing and electron transfer: Some insights from molecular modeling. *Electrochimica Acta* **508**, 145182 (2024).
 4. Sampei, H. *et al.* Factors governing the protonation of kegggin-type polyoxometalates: Influence of the core structure in clusters. *Dalton Transactions* **53**, 8576–8583 (2024).
 5. Han, M. *et al.* Emerging polyoxometalate clusters-based redox flow batteries: Performance metrics, application prospects, and development strategies. *Energy Storage Materials* **71**, 103576 (2024).
 6. Nielsen, M. T., Moltved, K. A. & Kepp, K. P. Electron transfer of hydrated transition-metal ions and the electronic state of Co³⁺(aq). *Inorg. Chem.* **57**, 7914–7924 (2018).
 7. Yang, L. *et al.* Tuning the inner- and outer-sphere electron transfer behavior of aqueous {CoW₁₂} polyoxometalate clusters for redox flow batteries exceeding 1.5V. *Energy Storage Materials* **65**, 103149 (2024).
 8. Guo, S.-X., Mariotti, A. W. A., Schlipf, C., Bond, A. M. & Wedd, A. G. A Systematic approach to the simulation of the voltammetric reduction of [α-SiW₁₂O₄₀]⁴⁻ in buffered aqueous electrolyte media and acetonitrile. *Journal of Electroanalytical Chemistry* **591**, 7–18 (2006).

Reviewer's Comments

Reviewer #1 (Remarks to the Author):

The authors have properly addressed my comments and the revised manuscript can be published as is.

Reply: Thank you for your positive comments.

Reviewer #2 (Remarks to the Author):

I think the paper is now ready for publication.

Reply: Thank you for your positive comments.

Reviewer #3 (Remarks to the Author):

The authors introduce a novel, low-potential anolyte active material in the form of Na-substituted phosphotungstic acid that provides full-cell voltages of aqueous RFBs of up to an OCV of 2.0 V.

The revised manuscript addresses all reviewer suggestions in depth. The authors took sufficient time to consider all the comments and adjusted the manuscript accordingly. However, during the revision process, more minor typos became evident, and I would like to request some additional information:

Reply: Thank you for your positive comments and professional suggestions.

- Page 20, Line 344: I thank the authors for providing the requested information. However, could you please briefly add the method used to obtain those results? Did you use gravimetric or UV-Vis measurements? If possible, also add the corresponding data to the supporting information.

Reply: Gravimetric measurements were applied to obtain the saturated concentration at 25 °C and 35°C. The details were provided in manuscript Page 22, line: 438-446.

Solubility Measurements. The weight method was used to measure the solubility of 3Na-PW₁₂ in anolyte. An excessive amount of 3Na-PW₁₂ was added into 30 ml of water, then heated to dissolve it. The solution was maintained in a water bath at 25 or 35 °C with stirring for 24 hours to obtain a saturated solution with 3Na-PW₁₂ precipitates. 10.00 ml of the clear saturated solution was quickly transferred into a beaker, and

completely dried at 120 °C overnight. Weighed the remaining 3Na-PW₁₂ solid, and it was noted as *m*. The saturated concentration of the 3Na-PW₁₂ solution at 25 or 35 °C was calculated by using the equation:

$$C_{\text{Saturated}} = m/0.01 \text{ (g l}^{-1}\text{)} \quad (9)$$

- Page 20, Line 349: Do the authors mean “spectra” instead of “spectroscopy”?

Reply: Thank you very much for your help. The mistake mentioned was corrected.

- Page 23, Line 406: I would suggest a rephrasing of “was calculated in the CV measurements with different scan rates” to “was applied to the CV measurement row at different scan rates”.

Reply: Thank you very much for your help. The mistake mentioned was corrected.

- Page 27, Line 476: Do the authors mean “worked” instead of “worded”?

Reply: The mistake mentioned was corrected.